# Stochastic Collapse: How Gradient Noise Attracts SGD Dynamics Towards Simpler Subnetworks

**Feng Chen**[*]   **Daniel Kunin**[*]   **Atsushi Yamamura** (山村篤志)[*]   **Surya Ganguli**

Stanford University

{fengc,kunin,atsushi3,sganguli}@stanford.edu

## Abstract

In this work, we reveal a strong implicit bias of stochastic gradient descent (SGD) that drives overly expressive networks to much simpler subnetworks, thereby dramatically reducing the number of independent parameters, and improving generalization. To reveal this bias, we identify *invariant sets*, or subsets of parameter space that remain unmodified by SGD. We focus on two classes of invariant sets that correspond to simpler (sparse or low-rank) subnetworks and commonly appear in modern architectures. Our analysis uncovers that SGD exhibits a property of *stochastic attractivity* towards these simpler invariant sets. We establish a sufficient condition for stochastic attractivity based on a competition between the loss landscape's curvature around the invariant set and the noise introduced by stochastic gradients. Remarkably, we find that an increased level of noise strengthens attractivity, leading to the emergence of attractive invariant sets associated with saddle-points or local maxima of the train loss. We observe empirically the existence of attractive invariant sets in trained deep neural networks, implying that SGD dynamics often collapses to simple subnetworks with either vanishing or redundant neurons. We further demonstrate how this simplifying process of *stochastic collapse* benefits generalization in a linear teacher-student framework. Finally, through this analysis, we mechanistically explain why early training with large learning rates for extended periods benefits subsequent generalization.

## 1  Introduction

The remarkable performance of modern deep learning systems relies on a complex interplay between a training dataset, a network's architecture, and an optimization strategy. Contrary to traditional statistical learning theory, these highly expressive models exhibit impressive generalization capabilities on natural tasks, even without explicit regularization, despite having the capacity to memorize random data [1]. This phenomenon is often attributed to implicit biases introduced in the training process that drive the learning dynamics towards models with low-complexity, thereby improving generalization. It is widely believed that a central source of this implicit bias is the randomness introduced by stochastic gradient descent (SGD) [2]. In this work we discuss SGD's role in attracting the dynamics, throughout training, towards subsets of parameter space that correspond to simpler subnetworks. We reveal that the architecture of modern deep neural networks as well as the nonlinear activation function plays a crucial role in forming these subsets, thus providing a novel perspective on the source of SGD's implicit bias. Our contributions are as follows:

1. We introduce *invariant sets* as subsets of parameter space that, once entered, trap SGD. We characterize two such sets that correspond to simpler subnetworks and appear extensively in modern architectures: one for vanishing neurons and the other for identical neurons (Sec. 3).

---

[*]Equal contribution. Ordered alphabetically.

2. We formulate a sufficient condition for *stochastic attractivity* — a process attracting SGD dynamics towards invariant sets — that reveals a competition between the loss landscape's curvature around an invariant set and the noise introduced by stochastic gradients (Sec. 4).

3. We apply the attractivity condition to determine when neurons with origin-passing activation functions collapse their parameters to zero, effectively removing the neuron. We empirically show that the parameters of permutable neurons within the same hidden-layer can collapse towards invariant sets corresponding to identical neurons (Sec. 5).

4. We demonstrate how this process of *stochastic collapse* influences generalization in a linear teacher-student framework. We apply these findings to shed light on the empirically observed importance of maintaining a large learning rate for an extended period during the early stages of training (Sec. 6).

## 2 Related Work

**Implicit biases of SGD.** Several recent works have explored the properties of SGD and its implicit biases. Barrett and Dherin [3] showed how SGD can be interpreted as adding implicit gradient norm regularization and Geiping et al. [4] showed that training full-batch gradient descent while explicitly adding this regularization can achieve the same performance as SGD. Kunin et al. [5] showed how the anisotropic structure of SGD noise effectively modifies the loss and HaoChen et al. [6] showed that parameter-dependent noise introduces an implicit bias towards local minima with smaller variance, while spherical Gaussian noise does not. Blanc et al. [7] and Damian et al. [8] studied the dynamics of gradient descent with label noise near a manifold of minima and proved that SGD implicitly regularizes the trace of the Hessian. Li et al. [9] developed a framework to study this bias of SGD by considering the projection of the trajectory to the manifold. Kleinberg et al. [10] showed how SGD is effectively operating on a smoother version of the original loss allowing it to escape sharp local minima. Zhu et al. [11] further demonstrated how the anisotropic structure of SGD noise helps SGD escape efficiently. Xie et al. [12] demonstrated that the covariance matrix of SGD approximates the Hessian around local minima, and thus the escape rate from a minima is linked to the Hessian eigenvalues or flatness. While numerous studies have sought to identify the origin of SGD's implicit bias, most have predominantly focused on how SGD introduces an implicit regularization term [3–5], minimizes a measure of curvature among equivalent minima [7–9], or escapes from sharp local minima [10–13]. In our work, we provide a novel perspective and discuss how stochastic gradients introduce a strong attraction towards regions of parameter space associated with simpler subnetworks, even when this attraction is detrimental to the full-batch train loss.

**Simplicity biases.** Several works have explored implicit biases that encourage some notion of sparsity or low-rankness in neural network learning dynamics. Kunin et al. [14] showed how gradient flow training with an exponential loss can lead to sparse solutions via an implicit max-margin process. Woodworth et al. [15] demonstrated how an implicit $L_1$ penalty occurs for diagonal linear networks trained with gradient flow in the limit of small initialization. Nacson et al. [16] and Pesme et al. [17] showed how large step sizes and stochastic gradients further bias these networks towards the sparse regime. Andriushchenko et al. [18] observed that longer training with large learning rates keeps SGD high in the loss landscape where an implicit bias towards sparsity is stronger and theoretically analyzed diagonal linear networks, showing that an associated SDE has implicit bias towards sparser representations. Vivien et al. [19] studied the role of label noise in inducing an implicit Lasso regularization for quadratically parameterized models. Kunin et al. [20] and Ziyin et al. [21] showed how weight decay induce a bias towards rank minimization for linear networks. Galanti et al. [22] and Wang and Jacot [23] extended this observation to deep settings trained with SGD. Jacot [24] theoretically, and Huh et al. [25] empirically, showed an implicit bias towards learning low-rank functions with increasing depth. Gur-Ari et al. [26] showed that gradients of SGD converge to a small subspace spanned by a few top eigenvectors of the Hessian. Ziyin et al. [27] demonstrated how data augmentation can promote dimensional collapse of representations in self-supervised learning. While many works have explored the emergence of sparsity or low-rankness during training, much of the analysis is confined to a particular architecture or consists of general empirical observations lacking a clear underlying mechanism. In our work, we focus on a fundamental characteristic of neural networks trained with SGD that leads to novel empirical observations.

Our analysis is closely related to a recent work studying failure modes of SGD [28] and a concurrent follow up work studying the stability of SGD near a fixed point [29]. In App. A we further discuss these works and other relevant works.

# 3   Invariant Sets of SGD Generated by Reflection Symmetries

Throughout this work, we consider a feed-forward network[2],

$$f_\theta(x) = w^{(m)}\sigma\left(w^{(m-1)}\cdots\sigma\left(w^{(1)}x + b^{(1)}\right)\cdots + b^{(m-1)}\right) + b^{(m)},\qquad(1)$$

parameterized by $\theta = (w^{(1)}, b^{(1)}, \ldots, w^{(m-1)}, b^{(m-1)}, w^{(m)}, b^{(m)}) \in \mathbb{R}^d$, where $w$ and $b$ denote weights and biases respectively, the superscript indexes the $m$ layers, and $\sigma(\cdot)$ is an activation function. The network is trained by stochastic gradient descent (SGD) over a training dataset $\{(x_1, y_1), \ldots, (x_n, y_n)\}$ of size $n$, where $x_i \in \mathbb{R}^p$ and $y_i \in \mathbb{R}^c$. The network parameters are randomly initialized at $\theta^{(0)}$ and iteratively updated according to

$$\theta^{(t+1)} = \theta^{(t)} - \frac{\eta}{\beta}\sum_{i \in \mathcal{B}^{(t)}}\nabla_\theta\ell\left(\theta^{(t)}; x_i, y_i\right),\qquad(2)$$

where $\eta > 0$ is the learning rate, $\mathcal{B}^{(t)} \subseteq [n]$ is a random[3] index set (mini-batch) of size $\beta$ at step $t$, and $\ell(\theta; x_i, y_i)$ is a loss function. We let $\mathcal{L}(\theta) = \frac{1}{n}\sum_{i \in [n]}\ell(\theta; x_i, y_i)$ denote the train loss averaged over the entire training dataset. Despite the simplicity of SGD's optimization procedure, understanding how it can navigate through complex, high-dimensional, non-convex loss landscapes to find generalizing solutions is still a mystery. Here, we take a dynamical systems perspective and identify subspaces of the parameters that are preserved under the SGD update equation (Eq. 2).

**Definition 3.1** (Invariant Set of SGD). *A Borel-measurable set $A \subseteq \mathbb{R}^d$ is an invariant set of SGD if given any initialization $\theta^{(0)} \in A$, all future iterates of SGD $\theta^{(t)}$ for $t \geq 0$ are contained within $A$ almost surely, for any batch size $\beta \in \mathbb{N}$, learning rate $\eta > 0$, and mini-batches $\{\mathcal{B}^{(t)} \subseteq [n] : t \in \mathbb{N}\}$.*

From this definition, it is immediately clear that all of parameter space $A = \mathbb{R}^d$ and any interpolating point $A = \{\theta_*\}$ such that $\mathcal{L}(\theta_*) = 0$ are invariant sets of dimension $d$ and $0$ respectively. However, the highly over-parameterized and layer-wise structure of neural network architectures lead to many additional, non-trivial invariant sets. We will focus on two fundamental invariant sets that appear ubiquitously in neural networks and correspond to simpler (sparse or low-rank) subnetworks.

**Proposition 3.1** (Sign Invariant Sets). *Consider a hidden neuron $p$ within layer $l$ of a feed-forward neural network. Let $(w_{in,p}^{(l)}, b_p^{(l)})$ and $w_{out,p}^{(l+1)}$ denote the parameters directly incoming and outgoing from the neuron respectively[4]. If the non-linearity $\sigma$ is origin-passing ($\sigma(0) = 0$), then the axial subspace $A = \{\theta \in \mathbb{R}^d | w_{in,p}^{(l)} = 0, b_p^{(l)} = 0, w_{out,p}^{(l+1)} = 0\}$ is an invariant set.*

The subspace corresponding to a sign invariant set represents the parameter space of a sparse subnetwork obtained by removing a hidden neuron. Most modern neural network architectures employ origin-passing activation functions, which includes linear, hyperbolic tangent, Rectified Linear Unit (ReLU) [30], Leaky ReLU [31], Exponential Linear Unit (ELU) [32], Swish [33], Sigmoid Linear Unit (SiLU), and Gaussian Error Linear Unit (GELU) [34]. Networks with any of these functions will exhibit invariant sets of this nature for each hidden neuron. A fully-connected network of depth $m$ and width $k$ possesses $(m-1)k$ distinct sign invariant sets.

**Proposition 3.2** (Permutation Invariant Sets). *Consider two hidden neurons $p, q$ within the same layer $l$ of a feed-forward neural network. Let $(w_{in,p}^{(l)}, b_p^{(l)}), (w_{in,q}^{(l)}, b_q^{(l)})$ denote the parameters directly incoming to the neurons, and $w_{out,p}^{(l+1)}, w_{out,q}^{(l+1)}$ the parameters directly outgoing from the neurons. The affine subspace $A = \{\theta \in \mathbb{R}^d | w_{in,p}^{(l)} = w_{in,q}^{(l)}, b_p^{(l)} = b_q^{(l)}, w_{out,p}^{(l+1)} = w_{out,q}^{(l+1)}\}$ is an invariant set.*

The subspace corresponding to a permutation invariant set represents the parameter space of a low-rank subnetwork obtained by constraining two neurons to be identical. All feed-forward networks possess this permutation invariance among the neurons in each hidden-layer. For a fully-connected network of depth $m$ and width $k$ there are $(m-1)\binom{k}{2}$ distinct permutation invariant sets.

**Invariant sets generated by symmetry.** The presence of sign and permutation invariant sets within neural networks is a direct result of reflection symmetries inherent in their architectural design. In this

---

[2]This notation encompasses fully-connected and convolutional networks, excluding more complex architectures such as transformers for simplicity, although much of our theory would directly apply.

[3]For simplicity we assume sampling with replacement such that there is no dependency between batches.

[4]The incoming and outgoing weights are related by: $(w_{in,p}^{(l)})_q = (w_{out,q}^{(l-1)})_p$.

context, a *symmetry* is defined as any transformation of the parameter that preserves the network's input-output mapping, regardless of the input. As stated in the following theorem, any approximate linear symmetry defined by a symmetric or orthogonal matrix generates an invariant set:

**Theorem 3.1** (Symmetry Induced Invariant Sets). *Let $Q \in \mathbb{R}^{d \times d}$ be a symmetric ($Q = Q^\intercal$) or orthogonal matrix ($QQ^\intercal = I_d$). The affine subspace $A = \{\theta \in \mathbb{R}^d | Q\theta = \theta\}$ is an invariant set if the loss function $\ell(\theta; x_i, y_i)$, for any $i \in [n]$, is approximately $Q$-symmetric around $A$, i.e., for any $\epsilon > 0$, there exists $\delta > 0$ such that $|\ell(Q\theta; x_i, y_i) - \ell(\theta; x_i, y_i)| < \epsilon d(\theta, A)$ for any $\theta \in \mathbb{R}^d$ satisfying $d(\theta, A) < \delta$ where $d(\theta, A)$ is the Euclidian distance between $\theta$ and $A$[5].*

Remarkably, the sign and permutation invariant sets defined earlier can be understood through this symmetry perspective. In both cases, the symmetry is defined by a symmetric and orthogonal matrix. This class of matrices describes the generalized notion of a reflection in $d$-dimensional space and the invariant sets are the axes of reflection, which can be at most $(d-1)$-dimensional. In the case of permutation invariant sets, any two hidden neurons in the same layer of a network can be permuted without changing the input-output function of the network, and thus the affine subspace $A$ where these two neurons are identical is an invariant set. On the other hand, in the case of sign invariant sets, the symmetric transformation $Q$ corresponds to a simultaneous sign flip of a neuron's input/output weights. This transformation does not change the input-output function of the network, as long as the activation function is odd. However, even for origin-passing activation functions that are not exactly odd, the loss function is approximately $Q$-symmetric around $A$, since the activation function is approximately odd around its origin. Therefore, Theorem 3.1 implies that the axial subspace $A$, where input and output weights are exactly zero, is an invariant set.

**Additional invariant sets.** In App. B, we discuss other classes of invariant sets, associated with softmax nonlinearities, low dimensional structure in data, and provide a further discussion on the connection between symmetry and invariant sets. We also discuss how our definition of invariant sets relates to other works [35–38] where symmetry and invariance are used to explore geometric properties of the loss landscape and the presence of critical points.

## 4   Gradient Noise Attracts SGD Dynamics towards Invariant Sets

To study the dynamics of SGD we will approximate[6] its trajectory with a stochastic differential equation (SDE), a common analysis technique applied in many works [5, 39–43]. SGD can be interpreted as full-batch gradient descent plus a per-step gradient noise term introduced by the random minibatch. Let $\xi_i(\theta)$ represent the per-sample gradient noise at $\theta$. It is easy to check that $\mathbb{E}[\xi_i(\theta)] = 0$ for all $\theta$. This motivates studying the SDE, which we refer to as *stochastic gradient flow (SGF)*,

$$d\theta_t = -\nabla\mathcal{L}(\theta_t)dt + \sqrt{\frac{\eta}{\beta}}\Sigma(\theta_t)dB_t, \qquad \theta_0 = \theta^{(0)}. \tag{3}$$

Here $B_t$ denotes a standard $d$-dimensional Wiener process and $\Sigma(\theta_t) = \sqrt{\mathbb{E}[\xi_i(\theta)\xi_i(\theta)^\intercal]}$. The drift term of this SDE is the negative full-batch gradient $-\nabla\mathcal{L}(\theta_t)$, while the diffusion is determined by a spatially-dependent diffusion matrix $D(\theta_t) = \frac{\eta}{2\beta}\Sigma(\theta_t)\Sigma(\theta_t)^\intercal$. Although this SDE cannot be interpreted as the continuous-time limit of SGD, it matches the first and second moments of the SGD process. See Appendix D for further discussions on the relationship between SGD and SGF. Additionally, SGF preserves all affine invariant sets of SGD:

**Proposition 4.1** (Informal). *All affine invariant sets of SGD are also affine invariant sets of SGF.*[7]

Like deterministic dynamical processes, the stochastic processes of SGF can be attracted toward the invariant sets. To describe this attraction, we adopt the concept from stochastic control theory [44].

**Definition 4.1** (Stochastic Attractivity). *An invariant set $A \subset \mathbb{R}^d$ of a stochastic process $\{\theta_t \in \mathbb{R}^d : t \geq 0\}$ is stochastically attractive[8] if for any $\rho > 0$ and $\epsilon > 0$, there exists $\delta > 0$ such that for any $\theta \in \mathbb{R}^d$ with the Euclidean distance $d(\theta, A) < \delta$,*

$$\mathbb{P}\left[\sup_{t \geq 0} d(\theta_t, A) \geq \epsilon \middle| \theta_0 = \theta\right] \leq \rho. \tag{4}$$

---

[5]The Euclidian distance between $\theta$ and $A$ is defined as $d(\theta, A) := \min_{a \in A} \|\theta - a\|$.

[6]As discussed in Li et al. [39], Eq. 3 is an order 1 weak approximation of Eq. 2.

[7]See App. C for the definition of invariant sets for continuous processes and the formal statement.

[8]This property is called stochastic stability in [44].

Intuitively, a set is stochastically attractive if there always exists a close initial condition that guarantees all future iterates remain close to the set with high probability. Understanding when the dynamics are attracted by a particular invariant set becomes fundamental for understanding the implicit bias of SGD. To gain insight into this process lets consider a canonical example in one-dimension.

**Geometric Brownian motion.** Geometric Brownian Motion (GBM) is a linear stochastic process given by the SDE, $d\theta_t = \mu\theta_t dt + \zeta\theta_t dB_t$, where $\mu \in \mathbb{R}$ represents the drift rate, $\zeta \in \mathbb{R}$ the volatility rate. Denote $\theta_0 \in \mathbb{R}$ as the initialization. The set $A = \{0\}$ is an invariant set of GBM, which, depending on the relationship between $\mu$ and $\zeta$, can be stochastically attractive. As the trajectory of GBM approaches the invariant set, its movement diminishes. To account for this effect, we take the logarithmic transformation $z_t = \log(\theta_t)$, where we assume without loss of generality that $\theta_0 > 0$. The stochastic process $z_t$ obeys the transformed SDE, $dz_t = \left(\mu - \zeta^2/2\right) dt + \zeta dB_t$, which has an additional drift term $-\zeta^2/2$ from Itô's lemma that attracts the process towards the invariant set. The total drift is now determined by a competition between the original drift rate $\mu$ and the additional attractive force driven by the diffusion. This competition controls whether the invariant set $A = \{0\}$ of GBM is stochastically attractive (i.e. when $\mu < \zeta^2/2$ the invariant set is attractive).

**Stochastic attraction in one-dimension.** Stochastic attractivity is a local property and thus we might expect SGF to act like GBM around an invariant set, enabling us to derive broader conclusions about stochastic attraction. If we consider SGF with a smooth loss and diffusion in one-dimension, where without loss of generality we assume $A = \{0\}$ is an invariant set, we can Taylor expand the gradient and the diffusion around the set, yielding geometric Brownian motion near the set:

$$d\theta_t \approx -\mathcal{L}''(0)\theta_t dt + \sqrt{D''(0)}\theta_t dB_t. \tag{5}$$

Here we use $\mathcal{L}'(0) = D(0) = D'(0) = 0$, which is a consequence of $\{0\}$ being an invariant set. From Eq. 5 we can formulate a necessary/sufficient condition for stochastic attractivity in one-dimension:

**Theorem 4.1** (Necessary/Sufficient Condition for Stochastic Attraction in One-Dimension). *Let $A = \{0\} \in \mathbb{R}$ be an invariant set of $\{\theta_t \in \mathbb{R} : t \geq 0\}$ obeying Eq. 3. Suppose $\mathcal{L}'(\theta) : \mathbb{R} \to \mathbb{R}, D(\theta) : \mathbb{R} \to \mathbb{R}$ are $C^2$ functions such that $D''(0) > 0$. Define rate of attractivity $\alpha = \mathcal{L}''(0) + \frac{1}{2}D''(0)$. A is stochastically attractive if $\alpha > 0$, while it is not stochastically attractive if $\alpha < 0$.* [9]

One of the most surprising implications of Theorem 4.1 is that SGF can potentially converge to a saddle-point or local maxima of a loss landscape, an observation previously made by Ziyin et al. [28]. To see this, recall that $D(\theta) \geq 0$ by definition, and $D(0) = 0$ as $\{0\}$ is an invariant set. As a result, $D'(0) = 0$ and $D''(0) \geq 0$ by the continuity assumption of $D$. The collapsing condition, therefore crucially depends on the curvature of the loss function $\mathcal{L}$ at $\theta = 0$. When $D''(0)$ is strictly positive, the collapsing condition can still be satisfied with negative curvature provided that $\mathcal{L}''(0) > -\frac{1}{2}D''(0)$. Given that $D$ is proportional to $\eta/\beta$, the learning rate to batch size ratio determines the maximum attainable steepness for an invariant set to be attractive.

**An illustrative example.** Consider SGF in a double-well potential $\mathcal{L}(\theta) = \frac{1}{4}(\theta^2 - \mu)^2$ with multiplicative diffusion[10], such that the dynamics are $d\theta_t = -(\theta_t^3 - \mu\theta_t)dt + \zeta\theta_t dB_t$ where $\mu, \zeta > 0$. The minima of the potential are located at $\theta = \pm\sqrt{\mu}$, while $\theta = 0$ is a local maximum and forms an invariant set. This example is special as the dynamics have an analytical steady-state distribution $p_{ss}(\theta)$, given any initialization. Thus, determining the stochastic attractivity of $\{0\}$ can be achieved by examining the steady-state distribution. As in GBM, we assume, without loss of generality, that $\theta_0 > 0$ and consider the logarithm process $z_t = \log(\theta_t)$. In this new coordinate, the noise term is constant, which allows us to determine the steady-state distribution. Transforming back to the original coordinate, we find that $p_{ss}(\theta)$ is given by a Gibbs distribution $p_{ss}(\theta) \propto e^{-\kappa\Psi(\theta)}$ for a modified potential[11] $\Psi(\theta) = \theta^2/2 - (2\mu/\zeta^2 - 2)\log(\theta)$ with constant $\kappa = 2/\zeta^2$ and partition function $Z = \int_{-\infty}^{\infty} e^{-\kappa\Psi(\theta)}d\theta$. When $\mu \leq \zeta^2/2$, the partition function diverges and $p_{ss}(\theta)$ collapses to a Dirac delta distribution at $\theta = 0$. This transition agrees with the collapsing condition from Theorem 4.1.

---

[9]When $\alpha = 0$, higher-order derivatives of the loss and diffusion determine the attractivity of $A$.

[10]Here we build on a substantial literature, originating from the seminal work by Kramers [45], that investigates the escape rate between minima of a particle in a double-well potential with constant diffusion [46].

[11]The interpretation of a modified potential determined by the gradient noise that drives the learning dynamics was explored in [42] and [5]. Stochastic collapse is when the partition function for this modified loss diverges.

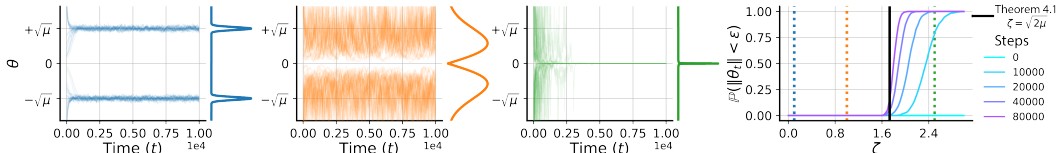

Figure 1: **Stochastic collapse in a quartic loss.** The left three plots show sample trajectories with the same, random initializations driven by the SDE $d\theta_t = -(\theta_t^3 - \mu\theta_t)dt + \zeta\theta_t dB_t$ for different values of $\zeta$. The analytic stationary solution is plotted on the side plot. **Leftmost:** for small $\zeta$ the steady-state distribution is two bumps around the minima $\theta = \pm\sqrt{\mu}$. **Middle left:** with increasing $\zeta$ the distribution spreads out and is biased towards $\theta = 0$. **Middle right:** when $\zeta$ surpasses the collapsing condition $\sqrt{2\mu}$ the steady-state distribution collapses to a dirac delta distribution at $\theta = 0$. **Rightmost:** the empirical probability of the sample trajectories at different steps being within $\epsilon$ of the origin as a function of increasing gradient noise. As we see empirically, there is a sudden phase transition which aligns with the collapsing condition from Theorem 4.1.

**Stochastic attraction in high-dimensions.** To extend the collapsing condition derived in Theorem 4.1 to high-dimensional cases, a natural idea would be to consider all one-dimensional slices of parameter space orthogonal to the invariant set. However, one challenge is that some of these slices might satisfy the collapsing condition while others do not. This can result in complex dynamics near the boundary of the invariant set making it difficult to derive a necessary and sufficient condition for attractivity in high-dimensions. Nonetheless, we can derive a sufficient condition:

**Theorem 4.2** (A Sufficient Condition for Stochastic Attraction in High-Dimensions)**.** *Let $A \subset \mathbb{R}^d$ be a $d_A$-dimensional affine subset, and a stochastic process $\{\theta_t \in \mathbb{R}^d : t \geq 0\}$ obey Eq. 3 in $A_c$, open c-neighborhood of $A$ with some $c > 0$. Suppose $\mathcal{L} : A_c \to \mathbb{R}$ is a $C^3$-function whose first and second-order derivatives are L-Lipschitz continuous. $D : A_c \to \mathbb{R}^{d \times d}$ is the diffusion matrix such that the second-order derivatives of its elements are L-Lipschitz continuous. Furthermore, we assume that all the elements of $\sqrt{D} : A_c \to \mathbb{R}^{d \times d}$ are L-Lipschitz continous. Let $D_\perp = P_\perp D P_\perp$ where $P_\perp : \mathbb{R}^d \to \mathbb{R}^d$ projects to the normal space of A. If there exists $\delta > 0$ such that $\nabla_{\hat{n}}^2 \hat{n}^\intercal D\hat{n}^\intercal > \delta$ and*

$$\nabla_{\hat{n}}^2 \left( \mathcal{L} - \frac{1}{2} \operatorname{Tr} D_\perp + (1-\delta)\hat{n}^\intercal D\hat{n} \right) > 0, \tag{6}$$

*for any unit normal vector $\hat{n} \in \mathbb{R}^d$ perpendicular to A and $\theta \in A$, then A is stochastically attractive.*

To prove Theorem 4.2 (see App. E), we consider a family of distance measures between $\theta_t$ and the invariant set $A$. Eq. 6 ensures that one of the distance measures is super-martingale in a neighborhood of the invariant set, allowing us to apply a version of Doob's maximal inequality (Lemma 1 in [44]).

## 5 Attractivity of Invariant Sets in Deep Neural Networks

In this section we explore theoretically and empirically the stochastic attractivity of invariant sets in deep neural networks.

**Sign invariant sets.** Sign invariant sets correspond to the parameter space of a sparse subnetwork obtained by removing a hidden neuron. Here, we demonstrate the stochastic collapse to a sign invariant set in a minimal model of neural networks—a scalar single neuron model: $f(x; w_1, w_2) = w_2\sigma(w_1x)$ with $w_1, w_2, x \in \mathbb{R}$. Suppose this model is trained via label-noise gradient descent and using the $L^2$-loss $\mathcal{L}(w_1, w_2) = \frac{1}{n}\sum_{i \in [n]}(y_i - f(x_i; w_1, w_2))^2$ on a non-empty data

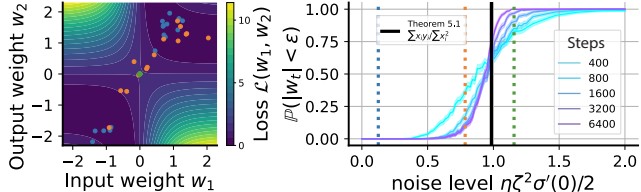

Figure 2: **Stochastic collapse of a single neuron.** The SDE dynamics of a single-neuron model are simulated with various noise levels $\zeta$. **Left:** A scatter map of trained weights $(w_1, w_2)$. The colors of the dots represent the noise level, corresponding to the vertical lines in the right panel. For each noise level, we plot 15 trained models with different noise realizations. The background heatmap shows the loss landscape. **Right:** The colored lines represent the empirical probability that $\sqrt{w_1^2 + w_2^2} < \epsilon = 10^{-2}$ (from $10^3$ samples) after a given number of update steps. We observe a sudden transition that aligns with the collapsing condition from Theorem 5.1.

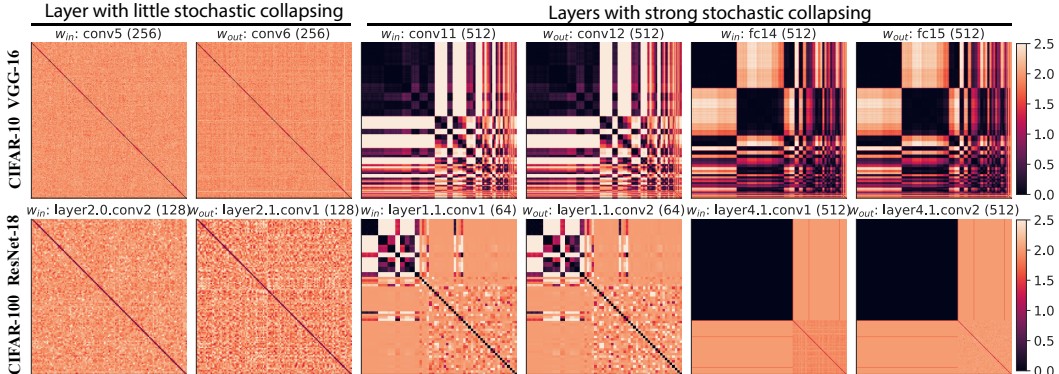

Figure 3: **Evidence of stochastic collapse towards permutation invariant sets in deep neural networks**. Each pair of plots shows the normalized distance matrix between incoming parameters ($w_{\text{in}}$) and outgoing parameters ($w_{\text{out}}$) of neurons within the same hidden layer. We show two pairs of plots with little stochastic collapsing and four pairs of plots with strong stochastic collapsing: three pairs for three different layers in VGG-16 trained for $10^5$ steps on CIFAR-10 (**top row**), and three pairs for three different layers in a ResNet-18 trained for $10^6$ steps on CIFAR-100 (**bottom row**). For each pair of plots, we sort the neurons by hierarchical clustering according to normalized distances between incoming parameters *only* (left plot in each pair), and then show the distance matrix between outgoing parameters using the *same* sorting of neurons (right plot in each pair). The similarity between each plot in a pair indicates that clusters of neurons with similar incoming parameters *also* have similar outgoing parameters. See App. J for experimental details.

set $\{(x_1, y_1), \cdots, (x_n, y_n)\}$ such that $\sum_{i \in [n]} x_i^2 \neq 0$. Exploiting Theorem 4.1, we can determine the attractivity of the sign invariant set $A = \{(0, 0)\}$ for this network.

**Theorem 5.1** (Informal). *Let $\{(w_{1,t}, w_{2,t}) \in \mathbb{R}^2 : t \geq 0\}$ be a process obeying label-noise gradient flow with $L^2$-loss of a single scalar neuron in a neighborhood $U$ of $A = \{(0, 0)\}$, with learning rate $\eta > 0$ and noise amplitude $\zeta > 0$. Suppose the activation function $\sigma : \mathbb{R} \to \mathbb{R}$ is smooth satisfying $\sigma(0) = 0$, $\sigma'(0) > 0$. The invariant set $A$ is stochastically attractive if $\frac{\sigma'(0)\eta\zeta^2}{2} > \frac{\left|\sum_{i \in [n]} x_i y_i\right|}{\sum_{i \in [n]} x_i^2}$.*

The term $\frac{\left|\sum_{i \in [n]} x_i y_i\right|}{\sum_{i \in [n]} x_i^2}$ can be thought of as the signal of the dataset. Thus, Theorem 5.1 states that stochastic attractivity is determined by balancing the signal and noise – an idea we will see later in Sec. 6 as the origin of generalization benefits from stochastic collapse. We leave the poof in App. F.

**Permutation invariant sets.** Permutation invariant sets correspond to subnetworks with fewer unique hidden neurons. An important question is to what extent are permutation invariant sets attractive – as this would indicate an implicit bias towards a low-rank model. We provide intuitive insights into the attractivity of permutation invariant sets through a toy example of a two-neuron neural network in App. G. More importantly, we empirically test whether permutation invariant sets can be attractive in deep neural networks via experiments using VGG-16 [47] and ResNet-18 [48] trained on CIFAR-10 and CIFAR-100 respectively [49]. Since the ReLU activation has additional symmetry resulting in a potentially larger invariant set [35], we replace ReLU with GELU activation in all our experiments below. To detect stochastic collapse to a permutation invariant set, we compute the normalized pairwise distance between parameters for neurons from the same layer, defined by $\text{dist}(w_i^{(l)}, w_j^{(l)}) \equiv \frac{2\|w_i^{(l)} - w_j^{(l)}\|_2^2}{\|w_i^{(l)}\|_2^2 + \|w_j^{(l)}\|_2^2}$. Strikingly, hierarchical clustering based on this distance reveals multiple large clusters of many neurons with near identical incoming and outgoing parameters after training (Fig. 3). Such clustering implies SGD implicitly drives weight matrices towards low-rank structures in many layers via attraction towards permutation invariant sets.

To explore the influence of the learning rate and batch size on the attraction towards sign and permutation invariant sets, we trained VGG-16 and ResNet-18 on CIFAR-10 and CIFAR-100, respectively, while varying these hyperparameters. We defined *vanishing* neurons as those with incoming and outgoing weights under 10% of the maximum norm for that layer, and identified the number of independent non-vanishing neurons by clustering the concatenated vector of incoming

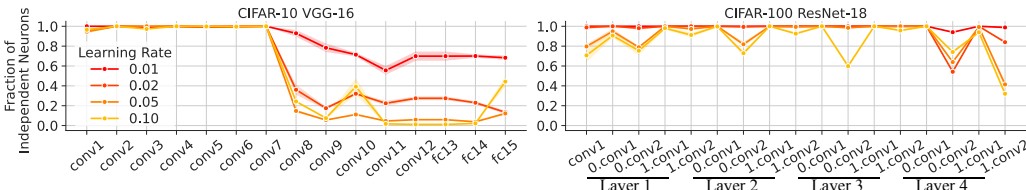

Figure 4: **Larger learning rates intensify stochastic collapse.** This figure illustrates how the fraction of independent neurons per layer in VGG-16 trained on CIFAR-10 (**left column**) and ResNet-18 trained on CIFAR-100 (**right column**) varies with changes in learning rates. The networks are evaluated at training steps of $10^5$. A reduced percentage of independent neurons indicates stronger stochastic collapse. See App. J for further details.

and outgoing parameters based on normalized distance. Two neurons were considered *identical* if their distance in parameter space was 10% of their norms, a stringent criterion given the high-dimensionality of the weight vectors.

We found that increased learning rates typically intensify the stochastic attraction to invariant sets by reducing the number of independent neurons (Fig. 4). A large reduction in the fraction of independent neurons was observed in VGG-16 between conv7 and conv8, where the number of channels increase from 256 to 512, indicating this excess model capacity is counteracted by stochastic collapse. Also, we note that Proposition 3.2 and 3.1 do not apply to neurons with residual connections. More strict definitions are required, as described in Proposition H.2 and H.1. We believe, this stricter constraint is the source of layer-to-layer oscillations in the fraction of independent neurons for ResNet18 in Fig. 4. Surprisingly, we noticed that unlike learning rate effects, changes in batch size produce complex shifts in the number of independent neurons. See Fig. 10 for a further discussion.

## 6 How Stochastic Collapse Can Improve Generalization

To understand how stochastic collapse can benefit generalization, we will theoretically analyze learning dynamics in a teacher-student model for linear networks, a well established framework for studying generalization [50–52]. This analysis not only provably connects the phenomenon of stochastic collapse to generalization benefits, but also suggests a stochastic collapse based mechanism for why successful learning rate schedules improve generalization. We then provide direct evidence for this theoretically suggested mechanism in deep neural networks.

**Insights from a two-layer linear neural network.** We build on the analysis of full-batch gradient learning dynamics of training error [53] and test error [51] in the limit of infinitesimal learning rates for two-layer linear neural networks in a student-teacher setting. In our new analysis here we incorporate the important new ingredient of stochastic gradients, which dramatically alters the learning dynamics. The details of our analysis are presented fully in App. I.

A teacher neural network with a low-rank weight matrix $\bar{W}$ generates a training dataset by providing noise corrupted outputs to a set of random Gaussian inputs. The input-output data covariance matrix $\Sigma_{yx}$ then drives the learning dynamics of a two-layer student with composite weight matrix $\hat{W} = W_2 W_1$. We analyze the learning dynamics of the student under a set of four assumptions precisely stated in App. I. These assumptions are similar to the ones made in [51, 53], and are roughly stated as: (A1) whitened inputs to the teacher; (A2) structured gradient noise related to input-output covariance; (A3) spectral initialization with student singular vectors aligned to input-output covariance singular vectors; (A4) balanced initialization with equal power in the two weight layers of the student. Under these assumptions, the learning dynamics of the student weight matrix $\hat{W} = W_2 W_1$ can be decomposed into independent learning dynamics for each singular value $\hat{s}_i$ of $\hat{W}$. An associated singular value $\tilde{s}_i$ of $\Sigma_{yx}$ drives the dynamics of $\hat{s}_i$ via the SDE,

$$d\hat{s}_i = 2\hat{s}_i \left( \left( \tilde{s}_i + \eta\zeta^2/2 \right) - \hat{s}_i \right) dt + 2\sqrt{\eta\zeta^2}\hat{s}_i dB_t, \tag{7}$$

where $\zeta > 0$ is the amplitude of the gradient noise. In the small noise limit of $\zeta \to 0$, this SDE aligns with the nonlinear ODE corresponding to the student network's dynamics under gradient flow, where exact solutions exist [51, 53]. Important insights gained from this previous analysis are that larger data singular modes are learned earlier [53] and larger data singular modes, when learned, help the student generalize, while smaller singular modes, when learned, impede generalization by

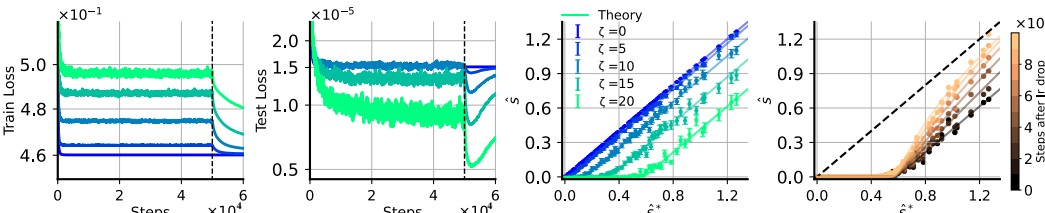

Figure 5: **Demonstrating generalization benefits of stochastic collapse in a teacher-student setting.** We show the train loss (**leftmost**) and test loss (**middle left**) during the training of the student with different gradient noises. Dashed lines in the leftmost and middle left panels indicate the step where learning rate is dropped. Training with larger gradient noises $\zeta$ generalizes better. **Middle right**: we show the noisy teacher signals against the learned student signals before dropping the learning rate. Larger noises have a stronger implicit bias towards zero. **Rightmost**: same as the middle right panel but we show the learned signals (for the brightest green in the middle right panel) at different training steps after the learning rate drop. All the results are averaged over 256 replicates.

overfitting to noise in the training data [51]. Both these insights will apply in our analysis of SGF. However, we will also uncover a third behavior unique to SGF: stochastic collapse effectively governs the minimum learnable singular mode, serving as a natural mechanism to prevent overfitting. Our analysis in App. I reveals that the student singular values $\hat{s}_i$ evolve according to the process:

**Theorem 6.1** (Stochastic Student Dynamics). *Under assumptions A1 - A4, the dynamics of $\hat{s}_i(t)$ given by Eq. 7 are governed by the stochastic process,*

$$\hat{s}_i(t) = e^{(2\tilde{s}_i - \eta\zeta^2)t + 2\sqrt{\eta\zeta^2}B_t} \left(2 \int_0^t e^{(2\tilde{s}_i - \eta\zeta^2)\tau + 2\sqrt{\eta\zeta^2}B_\tau} d\tau + \hat{s}_i(0)^{-1}\right)^{-1}. \tag{8}$$

In terms of the per layer balanced student weight $w_i = \sqrt{\hat{s}_i}$, the above process can be thought of as originating from the quartic loss landscape $\ell_i(w_i) = \frac{1}{4}(w_i^2 - \tilde{s}_i)^2$, equivalent to the example presented in Sec. 4. Thus, there exists an invariant set for each student singular value corresponding to the affine space $\hat{s}_i = w_i = 0$ where both drift and diffusion terms vanish. Stochastic collapse to this invariant set would thus induce a low-rank regularization on the student in which some number of singular modes in the data are *never learned* despite having nonzero data singular values $\tilde{s}_i$. To confirm stochastic collapse to this invariant set, we derive:

**Corollary 6.1.** *In the limit $\hat{s}_i(0) \to 0$, for any $t > 0$, then $\hat{s}_i(t)/\hat{s}_i(0) \underset{a.s.}{\to} e^{(2\tilde{s}_i - \eta\zeta^2)t + 2\sqrt{\eta\zeta^2}B_t}$.*

Corollary 6.1 has important implications. First, if $\tilde{s}_i < \frac{\eta\zeta^2}{2}$, the distribution will exhibit stochastic collapse, converging to a delta distribution at the origin, consistent with Theorem 4.1 (Fig. 5: middle right panel). This in turn implies that early training with large learning rates promotes stochastic collapse of more student singular modes. Indeed, even after the training loss plateaus, further extended training with a large learning rate $\eta$ will continue to drive stochastic collapse of small singular modes. After a subsequent learning rate drop, some of these modes will cease to obey the collapsing condition and as a consequence, the associated invariant set will repulse rather than attract the dynamics. Nonetheless, the repulsive escape will take longer the closer it starts from the invariant set (Fig. 5: rightmost panel). This results in dynamics in which generalization error is reduced by the student not learning many data singular modes with large $\eta$, because $\tilde{s}_i < \frac{\eta\zeta^2}{2}$, and then learning only the subset with large $\tilde{s}$ that have time to escape the vicinity of the invariant set after a learning rate drop. The overall effect is that the student selectively learns modes with higher signal-to-noise ratios, resulting in improved generalization after the learning rate drop as shown in Fig. 5: middle left panel. In summary, the linear teacher-student model highlights that with an appropriate learning rate schedule, a noisy student can effectively avoid overfitting through stochastic collapse and achieve superior generalization compared to a noiseless student.

**Stochastic collapse explains why early large learning rates helps generalization.** The analyses in the linear teacher student network provide valuable insights into how stochastic collapsing can enhance generalization. Importantly, it sheds light on the mystery of why training at large learning rates for a long period of time (even after training and test loss have plateaued) helps generalization after a subsequent learning rate drop. The key prediction is that a large learning rate induces stronger

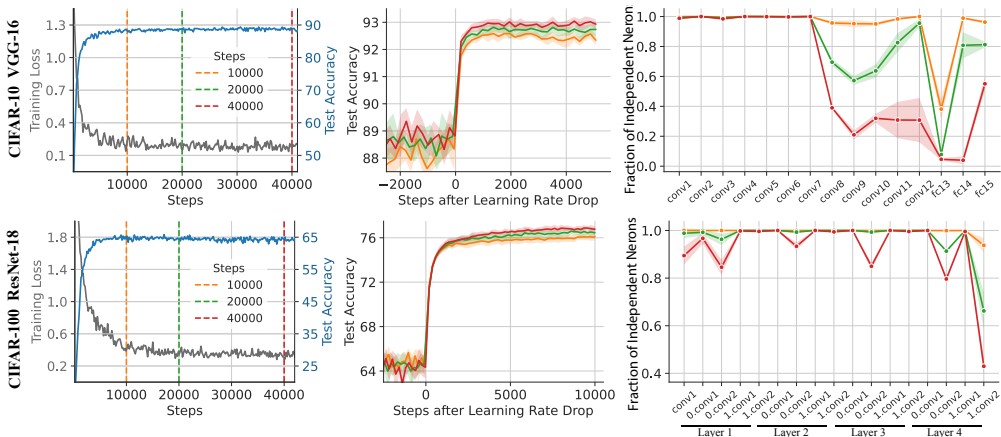

Figure 6: **Large learning rates aid generalization via stochastic collapse to simpler subnetworks.** **Left**: The training loss (grey) and test accuracy (blue) during an initial training phase using high learning rate (lr=0.1). Vertical dashed lines highlight the training steps at which we will drop the learning rate. **Middle**: Test accuracy preceding and following a learning rate drop to lr=0.01. Colors indicate when the learning rate drop shown in the left plot occured. Despite plateauing of the training loss and test accuracy, a later learning rate drop yields higher final test accuracy. **Right**: The fraction of independent neurons per layer evaluated at different learning rate drop times (indicated by the color) during the initial high learning rate training phase. Prolonged training with a large learning rate induces implicit regularization by reducing the number of independent neurons. All curves represent the average across eight replicates. See App. J for experimental details.

stochastic collapse, thereby regularizing the model complexity. Furthermore, remaining in a phase of larger learning rates for a prolonged period drives SGD closer to the invariant set. Consequently, when the learning rate is eventually dropped, overfitting in these specific directions is mitigated.

To test this predicted mechanism, we trained VGG-16 and ResNet-16 on CIFAR-10 and CIFAR-100, respectively. The training loss and test accuracy had already plateaued at $10^4$ steps (Fig. 6: left column). We dropped the learning rate after different lengths of the initial high learning rate training phase and confirmed that training with large learning rates for longer periods helps subsequent generalization (Fig. 6: middle column). We then tested our prediction that stochastic collapse is occurring during the early high learning rate training, thereby enhancing an implicit bias towards simpler subnetworks. In particular, we computed the fraction of independent neurons at each step where we drop the learning rate. We found, as predicted, that models with a longer initial training phase collapse towards invariant sets with fewer independent neurons (Fig. 6: right column).

## 7 Conclusion

In this study, we demonstrated how stochasticity in SGD biases overly expressive neural networks towards *invariant sets* that correspond to simpler subnetworks, resulting in improved generalization. We established a sufficient condition for *stochastic attractivity* in terms of the Hessian of the loss landscape and the noise introduced by stochastic gradients. We combined theory and empirics to study the invariant sets corresponding to vanishing neurons with origin-passing activation functions and identical neurons generated by permutation symmetry. Furthermore, we elucidated how this process of *stochastic collapse* can be beneficial for generalization using a linear teacher-student framework and explain the importance of large learning rates early in the training process.

**Limitations and future directions.** Activation function with additional continuous symmetries, such as ReLU, might have a larger class of invariant sets than those studied in our work, which remains to be explored. As all the invariant sets in our study are affine, exploring the possibility of curved invariant sets and how curvature affects the analysis is an interesting direction for future work. Furthermore, the interplay between symmetries in data and invariant sets warrants investigation. Identifying the necessary conditions for stochastic attractivity of general affine invariant sets is an important goal for future work. Extending our analytic results from the continuous SGF to the discrete SGD updates is an interesting theoretical direction. Lastly, designing new optimization algorithms based on our insights into stochastic collapse is a major goal for our future work.

## Acknowledgments and Disclosure of Funding

We thank Nan Cheng, Shaul Druckmann, Mason Kamb, Itamar Daniel Landau, Chao Ma, Nina Miolane, Mansheej Paul, Allan Raventós, Ben Sorscher, and Daniel Wennberg for helpful discussions. D.K. thanks the Open Philanthropy AI Fellowship for support. S.G. thanks the James S. McDonnell and Simons Foundations, NTT Research, and an NSF CAREER Award for support.

## Author Contributions

D.K. and F.C. initiated the project. F.C. formulated an initial analysis on stochastic collapse and is primarily responsible for the experiments associated with deep neural networks. D.K. is primarily responsible for the linear teacher-student analysis and the original observation that SGD can collapse to a saddle-point introduced by symmetry. A.Y. primarily contributed to the theoretical formulations and proofs, which include the concept of invariant sets and conditions for stochastic attractivity. S.G. advised throughout the work and provided funding for computation. All of the authors have worked on writing the manuscript.

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

# A  Extended Related Work

**SGD's noise covariance shape matters.** Our work is closely related to HaoChen et al. [6] work that showed how parameter-dependent noise introduces an implicit bias towards local minima with smaller variance, while spherical Gaussian noise does not. In particular, their work conjectured that the parameter dependent noise introduced by SGD has an implicit bias effect "toward parameters where the noise covariance is smaller". They studied this effect in two-layer quadratically-parameterized model introduced by [15]. Our work generalizes this effect to a broad range of models by introducing invariant sets and demonstrates that this implicit bias towards these regions is regulated by a competition between the curvature of the full-batch loss and the magnitude of the noise.

**SGD can converge to a local maximum.** Our work is closely related to a recent work studying failure modes of SGD [28] and a concurrent follow up work studying the stability of SGD near a fixed point [29]. Ziyin et al. [28] studied the behavior of SGD in one-dimensional quadratic and quartic losses and showed that SGD can converge to local maxima, escape from saddle-points slowly, and prefer sharp minima due to the gradient noise. A strength of their analysis is that they directly derive their results from discrete SGD updates, while we use a continuous formulation. However, their analysis is limited to one-dimensional scenarios and is used to illustrate behaviors of SGD they implicitly deemed unfavorable. In our work, we describe the mechanism of stochastic collapse generally and connect it to a simplicity bias that can benefit generalization. Concurrent to our work, Ziyin et al. [29] proposes a new notion of stability for SGD near a fixed point called probabilistic stability, which overlaps with our work in discussing stochastic collapse for single neurons but not for permutation invariance. They spotlight SGD learning phases using this stability concept, while we explore how stochastic gradients incline networks toward simpler subnetwork invariants.

**SGD learns sparse features.** Our work is closely related to Andriushchenko et al. [18] recent work studying a sparsity bias of SGD with large learning rates. The novelty of their work can be summarized as: 1. They revealed that large-step SGD dynamics have effective slow dynamics with multiplicative noise during loss stabilization. 2. They theoretically analyzed diagonal linear networks, showing that the SDE has implicit bias toward sparser representations. 3. They conjectured that deep nonlinear networks also show this phenomenon, which is supported by the empirical results. 4. They argued that SGD first learns sparse features and then fits the data after the step size is decreased. With this foundation laid, we would like to emphasize two fundamental aspects that set our work apart:

First, our analysis goes far beyond the basic understanding of diagonal linear networks, offering a broader perspective by introducing the invariant sets with a more general theorem (Theorem 4.2). In their work, the authors conjectured an implicit bias towards sparser features beyond their diagonal linear model, leaving this exploration open as an "exciting avenue for future work". We believe that our paper makes substantial progress along this avenue by understanding the implicit bias towards sparse representations as stochastic collapse to the invariant sets. Furthermore, our theory is applicable to general deep neural nets. We also provide a theoretical framework that predicts the quantitative conditions for this attractiveness in a general setting. While applying this theorem to arbitrary neural nets is a challenging future work, our work utilized this theorem to quantitatively analyze the collapsing threshold of simple models. Second, our results shed light on the collapsing phenomena of weight vectors, a perspective that contrasts with their emphasis on neuron similarity based on activation patterns. Given that weight vector similarities often lead to similarity in activation patterns, our findings suggest that neural networks adhere to a stronger condition.

# B  Further Discussion on Invariant Sets of SGD

In our work, we start by describing subsets of parameter space that are unmodified by SGD, which we term invariant sets. Several works have explored geometric properties of the loss landscape through the lens of symmetry and invariance. [35] demonstrated how the continuous symmetries of the loss function create level sets of the loss and impose geometric constraints on the gradients. [36] and [37] showed how permutation symmetries in the hidden neurons can lead to many potentially isolated minima with saddle points between them. [38] studied the learning dynamics of gradient flow near singular regions of parameter space, which for a multilayer perceptron, include permutation and sign invariance. A series of works [54–57] have leveraged the hierarchical structures of neural networks along with properties of their activation functions to identify critical points of the loss landscape.

## B.1  Proof of Proposition 3.1 and Proposition 3.2

*Proof.* $A$ is affine and $0 \in A$, thus it suffices to show that $\frac{\partial \ell}{\partial \theta} \in A$ and is independent of the training data. Denote the activation at layer $l$ as $h^{(l)} = \sigma\left(w^{(l)^T} h^{(l-1)} + b^{(l)}\right)$. The gradients associated with the outgoing weights are,

$$\frac{\partial \ell}{\partial w_{\text{out},i}^{(l+1)}} = \frac{\partial \ell}{\partial h^{(l+1)}} \odot \frac{\partial h^{(l+1)}}{\partial w_{\text{out},i}^{(l+1)}}$$

$$= \frac{\partial \ell}{\partial h^{(l+1)}} \odot \sigma'\left(w_{\text{out},i}^{(l+1)} h_i^{(l)} + b_i^{(l+1)}\right) \sigma\left(w_{\text{in},i}^{(l)^T} h^{(l-1)} + b_i^{(l)}\right). \tag{9}$$

If $\theta \in A$, $\frac{\partial \ell}{\partial w_{\text{out},p}^{(l+1)}} = 0$. The gradients associated with the incoming weights are,

$$\frac{\partial \ell}{\partial w_{\text{in},i}^{(l)}} = \left\langle \frac{\partial \ell}{\partial h^{(l+1)}}, \frac{\partial h^{(l+1)}}{\partial h_i^{(l)}} \right\rangle \frac{\partial h_i^{(l)}}{\partial w_{\text{out},i}^{(l)}}$$

$$= \left\langle \frac{\partial \ell}{\partial h^{(l+1)}}, \left(\sigma'\left(w_{\text{out},i}^{(l+1)} h_i^{(l)} + b^{(l+1)}\right) \odot w_{\text{out},i}^{(l+1)}\right) \right\rangle \sigma'\left(w_{\text{in},i}^{(l)^T} h^{(l-1)} + b_i^{(l)}\right) \odot h^{(l-1)}. \tag{10}$$

If $\theta \in A$, $\frac{\partial \ell}{\partial w_{\text{in},p}^{(l)}} = 0$. The gradients associated with the biases are,

$$\frac{\partial \ell}{\partial b_i^{(l)}} = \left\langle \frac{\partial \ell}{\partial h^{(l+1)}}, \frac{\partial h^{(l+1)}}{\partial h_i^{(l)}} \right\rangle \frac{\partial h_i^{(l)}}{\partial b_i^{(l)}}$$

$$= \left\langle \frac{\partial \ell}{\partial h^{(l+1)}}, \left(\sigma'\left(w_{\text{out},i}^{(l+1)} h_i^{(l)} + b^{(l+1)}\right) \odot w_{\text{out},i}^{(l+1)}\right) \right\rangle \sigma'\left(w_{\text{in},i}^{(l)^T} h^{(l-1)} + b_i^{(l)}\right). \tag{11}$$

If $\theta \in A$, $\frac{\partial \ell}{\partial b_p^{(l)}} = 0$. Therefore, $\frac{\partial \ell}{\partial \theta} \in A$. □

*Proof.* We can directly see from Eq. 9-11 that if $\theta \in A$, $\frac{\partial \ell}{\partial w_{\text{out},p}^{(l+1)}} = \frac{\partial \ell}{\partial w_{\text{out},q}^{(l+1)}}$, $\frac{\partial \ell}{\partial w_{\text{in},p}^{(l)}} = \frac{\partial \ell}{\partial w_{\text{in},q}^{(l)}}$ and $\frac{\partial \ell}{\partial b_p^{(l)}} = \frac{\partial \ell}{\partial b_q^{(l)}}$. Therefore, $\frac{\partial \ell}{\partial \theta} \in A$. □

## B.2  Proof of Theorem 3.1

*Proof.* It suffices to show that $\nabla \ell(\theta; x, y) \in A$ for all $\theta \in A$ and training data. In the following, we use the notation of $\ell(\theta)$ as a shorthand for $\ell(\theta; x, y)$. First, we show $\nabla \ell_Q(\theta) = \nabla \ell(\theta)$ for $\theta \in A$, where $\ell_Q(\theta) = \ell(Q\theta)$. The derivative of $\ell_Q$ along an arbitrary unit vector $n$ is given by

$$\nabla_n \ell_Q(\theta) = \lim_{h \to 0} \frac{\ell(Q(\theta + hn)) - \ell(Q\theta)}{h} = \lim_{h \to 0} \frac{\ell(\theta + hn) - \ell(\theta)}{h} + \frac{\ell(Q(\theta + hn)) - \ell(\theta + hn)}{h}.$$

If $n$ is a tangent vector of $A$, the second term is zero. If $n$ is a normal vector of $A$, then by the approximate $Q$-symmetry assumption, the second term converges to zero. These two facts imply that

$\nabla \ell_Q(\theta) = \nabla \ell(\theta)$. Then, for any $\theta \in A$, $\nabla \ell(\theta) = \nabla \ell_Q(\theta) = Q^\mathsf{T} \nabla \ell(Q\theta)$. Since $Q$ is symmetric $(Q = Q^\mathsf{T})$ or orthogonal $(QQ^\mathsf{T} = I_d)$, we obtain

$$Q \nabla \ell(\theta; x, y) = \nabla \ell(\theta; x, y),$$

which implies $\nabla \ell(\theta; x, y) \in A$. Hence, regardless of the sequence of mini-batches, starting from $\theta^{(0)} \in A$, the SGD trajectory remains within $A$, demonstrating $A$ is an invariant set. $\qquad\square$

Theorem 3.1 provides an alternative proof of Proposition 3.1 and Proposition 3.2, simply identify a symmetric or orthogonal matrix $Q$ that generates these sets. For the sign invariant sets, $Q$ is the identity matrix with negative diagonals restricted to the index of the parameters associated with the invariant set. For the permutation invariant set, $Q$ is the identity matrix with the diagonal blocks for the parameters associated with the two permutatable neurons swapped to being off diagonal.

## B.3   Additional Invariant Sets

Beyond the classes of invariant sets already presented, there can exist other invariant sets depending on the structure of the architecture. For example, for a network using a softmax layer, the columns of the weight matrix and the vector of biases immediately preceding this layer can be shifted by a constant without changing the outputs. As a result, the dynamics for these parameters are constrained to an affine space with a constant sum determined by their initialization. This subspace is an invariant set. Similarly, if the training data has a non-trivial kernel space, as in the case of overparameterized linear regression, then the rows of the first-layer's weight matrix can be translated by any vector in the kernel space without changing the output of the network. Again, this constrains the learning dynamics to an affine space, which is an invariant set. In general, any translational symmetry should generate invariant sets.

# C  Invariant sets of SGF

## C.1  Definition of invariant sets for continuous processes

Here, we provide the formal definition of invariant sets for a continuous process and a formal statement of Proposition 4.1.

**Definition C.1** (Invariant sets of continuous processes). *For a given stochastic process $\{\theta_t \in \mathbb{R}^d : t \geq 0\}$, a Borel-measurable set $A \subset \mathbb{R}^d$ is defined as invariant if for every initial point $\theta_0$ in A, the probability that the process stays in A for all $t \geq 0$ is 1, i. e.,*

$$\mathbb{P}[\theta_t \in A \text{ for any } t \geq 0 | \theta_0 \in A] = 1. \tag{12}$$

## C.2  Proof of Proposition 4.1

**Proposition 4.1** (Formal). *Consider an SGD process $\{\theta^{(t)} \in \mathbb{R}^d : t \in \mathbb{N}\}$ given by Eq. 2, where the gradients $\nabla\ell(\cdot; x_i, y_i) : \mathbb{R}^d \to \mathbb{R}^d$ in Eq. 2 are Lipschitz continuous and bounded for any $i \in [n]$. If $A \subseteq \mathbb{R}^d$ is an affine invariant set of SGD $\{\theta^{(t)} : t \in \mathbb{N}\}$, then A is also an invariant set of an SGF process $\{\theta_t : t \geq 0\}$ given by Eq. 3.*

*proof.* By assumption, $\nabla\ell(\theta; x_i, y_i)$ is $L$-Lipschitz in $\theta$ and $\|\nabla\ell(\theta; x_i, y_i)\| \leq L$ with some constant $L > 0$. Then we can see that $\nabla\mathcal{L}$ is also Lipschitz continuous and bounded as follows:

$$\|\nabla\mathcal{L}(\theta_1) - \nabla\mathcal{L}(\theta_2)\| = \frac{1}{n}\left\|\sum_{i \in [n]} (\nabla\ell(\theta_1; x_i, y_i) - \nabla\ell(\theta_2; x_i, y_i))\right\| \leq L\|\theta_1 - \theta_2\|$$

$$\|\nabla\mathcal{L}(\theta)\| \leq \frac{1}{n}\sum_i \|\ell(\theta; x_i, y_i)\| \leq L.$$

Let $\bar{\ell}(\theta, x_i, y_i) = \ell(\theta, x_i, y_i) - \mathcal{L}(\theta, x_i, y_i)$. $\nabla\bar{\ell}(\cdot, x_i, y_i)$ is clearly $2L$-Lipschitz continuous and is bounded by $2L$ for any $i \in [n]$. Then,

$$\|D_{lm}(\theta_1) - D_{lm}(\theta_2)\|$$

$$= \frac{1}{n}\left\|\sum_{i \in [n]} \left(\partial_l\bar{\ell}(\theta_1; x_i, y_i)\partial_m\bar{\ell}(\theta_1; x_i, y_i) - \partial_l\bar{\ell}(\theta_2; x_i, y_i)\partial_m\bar{\ell}(\theta_2; x_i, y_i)\right)\right\|$$

$$\leq \frac{1}{n}\sum_{i \in [n]} \left[2L(|\partial_l\bar{\ell}(\theta_1; x_i, y_i)| + |\partial_m\bar{\ell}(\theta_2; x_i, y_i)|)\|\theta_1 - \theta_2\|^2\right]$$

$$\leq 4L^2\|\theta_1 - \theta_2\|^2.$$

This implies that $\|D(\theta_1) - D(\theta_2)\|_\infty \leq 4L^2\|\theta_1 - \theta_2\|^2$, where $\|\cdot\|_\infty$ denotes the operator norm. On the other hand, the Powers-Stormer inequality gives,

$$\|\sqrt{D(\theta_1)} - \sqrt{D(\theta_2)}\|_F \leq \sqrt{d}\sqrt{\|D(\theta_1) - D(\theta_2)\|_1} \leq d\sqrt{\|D(\theta_1) - D(\theta_2)\|_\infty},$$

where $\|\cdot\|_F$ and $\|\cdot\|_1$ denote the Frobenius norm and Schatten 1-norm, respectively. Thus, $\sqrt{D(\theta)}$ is Lipschitz continuous with respect to the Frobenius norm. This establishes the existence of a unique strong solution to the SDE in Eq. 3. It now suffices to show that the SDE has a solution $\{\theta_t : t \geq 0\}$ satisfying Eq. 12. Let $P : \mathbb{R}^d \to \mathbb{R}^d$ be the projection operator to the affine subset $A$, and we construct a projected SDE:

$$d\tilde{\theta}_t = -P\nabla\mathcal{L}(\tilde{\theta}_t)dt + \sqrt{\frac{\eta}{\beta}}P\Sigma(\tilde{\theta}_t)dB_t.$$

By construction, this projected SDE also has a unique solution $\tilde{\theta}_t$, which clearly satisfies the condition Eq. 12. Moreover, given that $A$ is an invariant set of the original SGD process, we have $P\nabla\ell(\theta; x_i, y_i) = \nabla\ell(\theta; x_i, y_i)$ for all $i \in \mathcal{B}$ and $x \in A$. This means that $\{\tilde{\theta}_t : t \geq 0\}$ satisfies Eq.12 almost surely. Hence, we conclude that $A$ is also an invariant set of the solution to Eq.3. $\square$

## D  SDE formulations of learning dynamics with gradient noise

### D.1  The relationship between SGD and SGF

Recall the SGD update rule from Eq. 2.

$$\theta^{(t+1)} = \theta^{(t)} - \frac{\eta}{\beta} \sum_{i \in \mathcal{B}^{(t)}} \nabla_\theta \ell \left( \theta^{(t)}; x_i, y_i \right). \tag{13}$$

We factorize the right-hand side into the negative full-batch gradient and the deviation from the full-batch gradient,

$$\underbrace{\theta^{(t+1)} = \theta^{(t)} - \eta \nabla \mathcal{L}(\theta^{(t)})}_{\text{gradient descent}} - \sqrt{\eta} \underbrace{\left( \frac{\sqrt{\eta}}{\beta} \sum_{i \in \mathcal{B}} \nabla_\theta \ell \left( \theta^{(t)}; x_i, y_i \right) - \sqrt{\eta} \nabla \mathcal{L}(\theta^{(t)}) \right)}_{\text{gradient noise}}. \tag{14}$$

Let $\xi_\mathcal{B}(\theta)$ represent the gradient noise at $\theta$. Assuming we sample with replacement, the gradient noise can be disentangled into a sum of i.i.d. per-sample gradient noises: $\xi_\mathcal{B}(\theta) = \frac{\sqrt{\eta}}{\beta} \sum_{i \in \mathcal{B}} \xi_i$ where $\xi_i = \nabla \ell(\theta; x_i, y_i) - \nabla \mathcal{L}(\theta)$. It is easy to check that $\mathbb{E}[\xi_i(\theta)] = 0$ for all $\theta$. Assuming that the gradient noises are gaussian random variables, Eq. 14 turns into

$$\Delta \theta^{(t)} = -\eta \nabla \mathcal{L}(\theta^{(t)}) + \sqrt{2D(\theta^{(t)})} \Delta W, \tag{15}$$

where $\Delta W \sim \mathcal{N}(0, \eta I)$ and the diffusion matrix $D(\theta^{(t)}) = \frac{1}{2} \text{Var}(\xi_\mathcal{B}) = \frac{\eta}{2\beta} \text{Var}(\xi_i)$. Eq. 15 is a Euler-Maruyama discretization of the following continuous-time stochastic differential equation, which we denote as *stochastic gradient flow (SGF)*,

$$d\theta_t = -\nabla \mathcal{L}(\theta_t) dt + \sqrt{2D(\theta_t)} dB_t, \qquad \theta_0 = \theta^{(0)}. \tag{16}$$

The diffusion matrix can be expressed as:

$$D(\theta_t) = \frac{\eta}{2\beta} \left( \frac{1}{N} \sum_{i \in [N]} \left[ \nabla_\theta \ell \left( \theta_t; x_i, y_i \right) \nabla_\theta \ell \left( \theta_t; x_i, y_i \right)^T \right] - \nabla \mathcal{L}(\theta_t) \nabla \mathcal{L}(\theta_t)^T \right). \tag{17}$$

The diffusion matrix can be decomposed into a product of a parameter-independent magnitude term $D_m$ and a parameter-dependent shape term $D_s(\theta)$ such that $D(\theta) = D_m D_s(\theta)$, where

$$D_m = \frac{\eta}{2\beta} \tag{18}$$

$$D_s(\theta) = \frac{1}{N} \sum_{i \in [N]} \left[ \nabla_\theta \ell \left( \theta_t; x_i, y_i \right) \nabla_\theta \ell \left( \theta_t; x_i, y_i \right)^T \right] - \nabla \mathcal{L}(\theta_t) \nabla \mathcal{L}(\theta_t)^T \tag{19}$$

The optimization hyperparameters learning rate and batch size affect the magnitude term $D_m$, while the network architecture, the training data, and the loss function affect the shape term $D_s(\theta)$. For generic network architectures, determining the exact form of the shape term is quite complicated. For the purpose of analytical assessments, we sometimes pivot to considering label-noise gradient descent (LNGD) as an alternative to stochastic gradient descent (SGD).

### D.2  A simpler SDE formulation via Label-Noise Gradient Descent

In the following, we will restrict our discussion to the loss function with the mean square error. Our aim is to achieve analytical results by simplifying the diffusion matrix. In its most basic form, using the deterministic gradient descent, the diffusion matrix is simply 0. Expanding on this, what happens when we deliberately introduce noise into the gradients? One common method to achieve this is by adding noise to the dataset labels. If we introduce i.i.d. mean-zero, variance $\zeta^2$ Gaussian label noise per training label, denoted as $\xi_i$, it results in an update rule we refer to as label-noise gradient descent (LNGD),

$$\theta^{(t+1)} = \theta^{(t)} - \eta \nabla \mathcal{L}(\theta^{(t)}) - \sqrt{\eta} \frac{\sqrt{\eta}}{N} \sum_{i \in [N]} \nabla f_{\theta^{(t)}}(x_i) \xi_i. \tag{20}$$

This is a discretization of the following continuous-time stochastic differential equation, which we denote as *label-noise gradient flow (LNGF)*,

$$d\theta_t = -\eta \nabla \mathcal{L}(\theta_t) + \sqrt{2D_{LN}(\theta_t)} dB_t, \qquad \theta_0 = \theta^{(0)}. \tag{21}$$

where the diffusion matrix of LNGF can be expressed as,

$$D_{LN}(\theta_t) = \frac{\eta\zeta^2}{2} \frac{1}{N} \sum_{i \in [N]} \nabla f_{\theta_t}(x_i) \nabla f_{\theta_t}(x_i)^T, \tag{22}$$

which can also be decomposed into a magnitude term and a shape term,

$$D_m = \frac{\eta\zeta^2}{2} \tag{23}$$

$$D_s = \frac{1}{N} \sum_{i \in [N]} \nabla f_{\theta_t}(x_i) \nabla f_{\theta_t}(x_i)^T \tag{24}$$

Compared with SGF, the magnitude term here depends on the learning rate and the variance of the label noise. Compared with SGF, the shape term here is much simpler, which allows us to get analytical results in Sec. 5 and 6. To compare with the theory, we also conducted experiments with LNGD. We study label-noise in theory because it can replicate the implicit regularization effects of the mini-batch noise. Essentially, label noise serves as a foundational model that paves the way for a more comprehensive grasp of the more complex dynamics inherent to mini-batch noise.

# E Conditions for stochastic attractivity

## E.1 Proof of Theorem 4.2

Theorem 4.2 holds as a corollary of the following slightly more general statement where the drift term of the SDE is not necessarily a gradient of a function.

**Theorem E.1.** *Let $A \subset \mathbb{R}^d$ be a $d_A$-dimensional affine subset, and a stochastic process $\{\theta_t \in \mathbb{R}^d : t \geq 0\}$ obey the following SDE in $A_c$, open c-neighborhood of A with some $c > 0$:*

$$d\theta_t = \mu(\theta_t)dt + \sqrt{2D(\theta_t)}dW_t, \tag{25}$$

*where $\{W_t : t \geq 0\}$ is the d-dimensional standard Wiener process. Here $\mu : A_c \to \mathbb{R}$ is a L-Lipschitz $C^2$-function with its first-order derivatives being L-Lipschitz as well. $D : A_c \to \mathbb{R}^{m \times m}$ is the diffusion matrix such that the second-order derivatives of its elements are L-Lipschitz continuous. We further assume that all the elements of $\sqrt{D(\theta)}$ are L-Lipschitz continuous. Let $D_\perp = P_\perp D P_\perp$ where $P_\perp : \mathbb{R}^d \to \mathbb{R}^d$ projects to the normal space of A. Suppose A is an invariant set of $\theta_t$. A is stochastically attractive if there exists $\delta > 0$ such that for any unit normal vector $\hat{n} \in \mathbb{R}^d$ perpendicular to A and any $\theta \in A$,*

$$\nabla_{\hat{n}}^2 \hat{n}^\mathsf{T} D \hat{n} > \delta \tag{26}$$

$$\nabla_{\hat{n}} \hat{n}^\mathsf{T} \mu + \nabla_{\hat{n}}^2 \left( -\frac{1}{2} \operatorname{Tr} D_\perp + (1 - \delta)\hat{n}^\mathsf{T} D \hat{n} \right) > 0. \tag{27}$$

*Proof.* We exploit a local version of Doob's maximal inequality (Lemma. 1 in [44]). By the Lipschitz property of $\mu$ and $\sqrt{D}$, the SDE Eq. 25 has a unique strong solution up to the exit time $\tau_c = \inf\{t \geq 0 : \theta_t \notin A_c\}$. The stopped process $\{\tilde{\theta}_t = \theta_{t \wedge \tau_c} : t \geq 0\}$ is a continuous strong Markov process. It suffices to show that A is stochastically attractive for $\tilde{\theta}_t$. We first show that $\tilde{\mathcal{A}}l^\lambda(x) \leq 0$ for any $x \in A_\epsilon \backslash A$ with some $0 < \epsilon < c$, and $\lambda > 0$, where $\tilde{\mathcal{A}}$ is the infinitesimal operator of the stochastic process $\tilde{\theta}_t$ and $A_\epsilon$ denotes the open neighborhood of A within distance $\epsilon > 0$. We consider the Euclidian distance $l(\tilde{\theta}_t)$ between the process $\tilde{\theta}_t$ and the affine invariant set A.

The SDE for $l^\lambda(\tilde{\theta}_t)$ is given by,

$$\begin{aligned}
dl^\lambda(\tilde{\theta}_t) &= \left( -\lambda l^{\lambda-2}(\tilde{\theta}_t)\theta_t^\mathsf{T} P_\perp \mu + \lambda l^{\lambda-2}(\tilde{\theta}_t) \operatorname{Tr} D_\perp + \lambda(\lambda - 2)l^{\lambda-4}(\tilde{\theta}_t) \left( \theta_t^\mathsf{T} D_\perp \tilde{\theta}_t \right) \right) dt \\
&\quad + \lambda l^{\lambda-2}(\tilde{\theta}_t)\tilde{\theta}_t^\mathsf{T} \sqrt{2D_\perp} dW_t.
\end{aligned}$$

This means that for any $x \in A_c$,

$$\tilde{\mathcal{A}}l^\lambda(x) = -\lambda l^{\lambda-2}(x)x^\mathsf{T} P_\perp \mu + \lambda l^{\lambda-2}(x) \operatorname{Tr} D_\perp + \lambda(\lambda - 2)l^{\lambda-4}(x) (x^\mathsf{T} D_\perp x). \tag{28}$$

By the fact that A is an invariant set, $\tilde{\mathcal{A}}l^2(x) = 0$ holds for any $x \in A$. Therefore, we obtain $\operatorname{Tr} D_\perp(x) = 0$. Since $D_\perp$ is positive semi-definite, $D_\perp(x) = 0$ for any $x \in A$. Furthermore, since $\tilde{\mathcal{A}}f(x) = 0$ with $f(\theta_t) = P_\perp \theta_t$ for any $x \in A$, $P_\perp \mu(x) = 0$ for any $x \in A$.

Going back to Eq.28, we obtain

$$\tilde{\mathcal{A}}l^\lambda(x) = 2\lambda l^{\lambda-2}(x)G(x) \left( \frac{F(x)}{2G(x)} - 1 + \lambda/2 \right),$$

where

$$F(x) := \left( -x^\mathsf{T} P_\perp \mu(x) + \operatorname{Tr} D_\perp(x) \right), \qquad G(x) := \left( \frac{P_\perp x}{\|P_\perp x\|} \right)^\mathsf{T} D(x) \frac{P_\perp x}{\|P_\perp x\|}.$$

Notice that $G(x) > 0$ for any $x \in A_\epsilon \backslash A$ with small enough $\epsilon > 0$ because Eq. 26 holds and the second derivatives of $D(x)$ are L-Lipschitz. We want to show that there exists $\lambda > 0$ such that for any $x \in A_\epsilon \backslash A$, $\frac{F(x)}{2G(x)} - 1 + \lambda/2 < 0$ with some $\lambda > 0$. Parameterize $x$ by $x(\xi) = x_\| + \xi\hat{n}$, where $x_\| \in A$ and $\hat{n}$ is the unit normal vector perpendicular to A. From L'Hôpital's rule, we have,

$$\lim_{\xi \to 0} \frac{F(x_\| + \xi\hat{n})}{G(x_\| + \xi\hat{n})} = \frac{\nabla_{\hat{n}}^2 F(x_\|)}{\hat{n}^\mathsf{T} \nabla_{\hat{n}}^2 D(x_\|)\hat{n}}.$$

The second derivatives of $F$ can be calculated as follows:

$$\nabla_{\hat{n}}^2 F(x_{\parallel}) \quad = \quad -x_{\parallel}^{\mathsf{T}} P_{\perp} \nabla_{\hat{n}}^2 \mu - 2\nabla_{\hat{n}} \hat{n}^{\mathsf{T}} \mu + \nabla_{\hat{n}}^2 \operatorname{Tr} D_{\perp} = -2\nabla_{\hat{n}} \hat{n}^{\mathsf{T}} \mu + \nabla_{\hat{n}}^2 \operatorname{Tr} D_{\perp}. \quad (29)$$

Substituting this, we obtain

$$\lim_{\xi \to 0} \frac{F(x_{\parallel} + \xi\hat{n})}{2G(x_{\parallel} + \xi\hat{n})} = \frac{-2\nabla_{\hat{n}} \hat{n}^{\mathsf{T}} \mu(x_{\parallel}) + \nabla_{\hat{n}}^2 \operatorname{Tr} D_{\perp}(x_{\parallel})}{2\hat{n}^{\mathsf{T}} \nabla_{\hat{n}}^2 D(x_{\parallel})\hat{n}} < 1 - \delta. \quad (30)$$

By the assumed Lipschitz property of the derivatives of $\mu$ and $D$, $\nabla_{\hat{n}}^2 F(x)$ and $\nabla_{\hat{n}}^2 G(x)$ are $L$-Lipschitz in $A_\epsilon \backslash A$ with small enough $\epsilon > 0$. Therefore, for any $0 < \xi < \epsilon$,

$$\left| \nabla_{\hat{n}} \frac{F(x_{\parallel} + \xi\hat{n})}{G(x_{\parallel} + \xi\hat{n})} \right|$$

$$= \left| \frac{G\nabla_{\hat{n}} F - F\nabla_{\hat{n}} G}{G^2} \right|$$

$$\leq \frac{|(\nabla_{\hat{n}}^2 G(x) + L\epsilon)\epsilon^2 (\nabla_{\hat{n}}^2 F(x_{\parallel}) + L\epsilon)\epsilon - (\nabla_{\hat{n}}^2 F(x_{\parallel}) - L\epsilon)\epsilon^2 (\nabla_{\hat{n}}^2 G(x_{\parallel}) - L\epsilon)\epsilon|}{((\nabla_{\hat{n}}^2 G(x_{\parallel}) - L\epsilon)\epsilon^2)^2}$$

$$\leq \frac{|(\nabla_{\hat{n}}^2 G(x_{\parallel}) + L\epsilon)(\nabla_{\hat{n}}^2 F(x_{\parallel}) + L\epsilon) - (\nabla_{\hat{n}}^2 F(x_{\parallel}) - L\epsilon)(\nabla_{\hat{n}}^2 G(x_{\parallel}) - L\epsilon)|}{\epsilon(\nabla_{\hat{n}}^2 G(x_{\parallel}))^2}$$

$$\times \left( \frac{\nabla_{\hat{n}}^2 G(x_{\parallel})}{\nabla_{\hat{n}}^2 G(x_{\parallel}) - L\epsilon} \right)^2$$

$$\leq \epsilon^{-1} |(1 + \delta^{-1} L\epsilon)(1 - \delta + \delta^{-1} L\epsilon) - (1 - \delta - \delta^{-1} L\epsilon)(1 - \delta^{-1} L\epsilon)| \left(1 + (\delta - L\epsilon)^{-1} L\epsilon\right)^2$$

$$\leq L(3\delta^{-1} - 1)\left(1 + (\delta - L\epsilon)^{-1} L\epsilon\right)^2$$

$$\leq 2L(3\delta^{-1} - 1).$$

Therefore $F(x_{\parallel} + \xi\hat{n})/G(x_{\parallel} + \xi\hat{n})$ is $2L(3\delta^{-1} - 1)$-Lipschitz as a function of $\xi$ in $A_\epsilon \backslash A$. This implies that there exists $\epsilon > 0$ such that for any $x \in A_\epsilon \backslash A$, $F(x)/2G(x) < 1 - \delta/2$ and hence $\tilde{\mathcal{A}}(l^\delta(x)) \leq 0$. Therefore $l^\delta(x)$ is in the domain of $\tilde{\mathcal{A}}_{A_\epsilon}$ and $\tilde{\mathcal{A}}_{A_\epsilon}(l^\delta(x)) = \tilde{\mathcal{A}}(l^\delta(x)) \leq 0$, where $\tilde{\mathcal{A}}_{A_\epsilon}$ is the infinitesimal operator of the stopped process $\tilde{\theta}_{t \wedge \tau}$ with $\tau = \inf_{t \geq 0}\{l(\theta_t) \geq \epsilon\}$. By the local version of Doob's maximal inequality, for any $\epsilon' < \epsilon$,

$$\mathbb{P}[\sup_{t \geq 0} l^\delta(\tilde{\theta}_t) \geq \epsilon' | \tilde{\theta}_0 = x] = \mathbb{P}[\sup_{t \geq 0} l^\delta(\tilde{\theta}_{t \wedge \tau}) \geq \epsilon' | \tilde{\theta}_0 = x] \leq \frac{l^\delta(x)}{\epsilon'}.$$

Equivalently, for any $\epsilon' < \epsilon^{1/\delta}$,

$$\mathbb{P}[\sup_{t \geq 0} l(\tilde{\theta}_t) \geq \epsilon' | \theta_0 = x] \leq \frac{l^\delta(x)}{\epsilon'^{1/\delta}}.$$

By taking $l(x)$ to be arbitrarily small, we can bound the left-hand side with any positive value $\rho > 0$. This means that $A$ is stochastically attractive for $\tilde{\theta}_t$ and so is for $\theta_t$. $\qquad \square$

### E.2 Proof of Theorem 4.1

The sufficient condition is a direct corollary of Theorem 4.2, and thus we show the necessary condition here. By assumption, $\mathcal{L}''(x) + \frac{1}{2}D''(x) < -\delta$ at $x = 0$ with some $\delta > 0$. It suffices to show that there exists $\epsilon > 0$ such that the hitting time

$$\tau := \inf\{t \in \mathbb{R} : \theta_t \geq \epsilon\}$$

is finite almost surely for any $x \in \mathbb{R}$ with initialization $\theta_0 = x$.

We choose $\epsilon > 0$ small enough so that $D(x) > 0$ for any $x \in (-\epsilon, \epsilon)\backslash\{0\}$. Then we can consider another stochastic process $Y_t = 2^{-1/2} \int_{\epsilon/2}^{\theta_{t \wedge \tau}} D^{-1/2}(x)dx$, which follows the SDE below

$$dY_t = D^{-1/2}\left(-\mathcal{L}'(\theta_{t \wedge \tau}) - \frac{1}{2}D'(\theta_{t \wedge \tau})\right)dt + dW_t.$$

for any $0 \le t < \tau$. By further taking $\epsilon$ small,

$$D^{-1/2}\left(-\mathcal{L}'(\theta_{t\wedge\tau}) - \frac{1}{2}D'(\theta_{t\wedge\tau})\right) > \frac{\delta\epsilon}{2}D^{-1/2} \ge \frac{\delta\epsilon}{2}\Big/\sqrt{\sup_{0<x<\epsilon} D(x)} > \delta.$$

Thus, $Y_t > \delta t + Y_0 + W_t$. Therefore,

$$\tau = \inf\{t \in \mathbb{R} : Y_t \ge \frac{1}{\sqrt{2}}\int_{\epsilon/2}^{\epsilon} D^{-1/2}(x)dx\} \le \inf\{t \in \mathbb{R} : \delta t + W_t \ge \frac{1}{\sqrt{2}}\int_{\epsilon/2}^{\epsilon} D^{-1/2}(x)dx - Y_0\}.$$

Since $\delta > 0$, the right-hand side is almost surely finite with any value of $Y_0$. This immediately implies that $A$ is not stochastically attractive.

# F The attractivity of the sign invariant set of single scalar neuron

We here investigate when a neuron may collapse towards the sign invariant set. Consider a neural network consisting of only one neuron with scalar input and output: $f(x; w_1, w_2) = w_2\sigma(w_1 x)$, where $x, w_1, w_2 \in \mathbb{R}$ and $\sigma$ represents the activation function. We assume that $\sigma(x)$ is a smooth Lipschitz continuous function with $\sigma(0) = 0$. Let $\mathcal{D} = \{(x_1, y_1), \cdots, (x_n, y_n)\}$ denote the training data set where $x_i, y_i \in \mathbb{R}, i \in [n]$. We consider the loss function $\mathcal{L}$ given by

$$\mathcal{L}(w_1, w_2, t) = \frac{1}{n} \sum_{i \in [n]} (f(x_i; w_1, w_2) - y_i - \zeta_{i,t})^2,$$

where $\zeta_{i,t}$ is the label noise which we assume to be increments of the $n$-dimensional Wiener process. The gradient flow of the loss is given by

$$\frac{dw_1}{dt} = -\frac{\partial \mathcal{L}}{\partial w_1} = -\frac{1}{n} \sum_{i \in [n]} (f(x_i; w_1, w_2) - y_i) w_2 x \sigma'(w_1 x_i) - w_2 x \sigma'(w_1 x_i) \zeta_{i,t}$$

$$\frac{dw_2}{dt} = -\frac{\partial \mathcal{L}}{\partial w_2} = -\frac{1}{n} \sum_{i \in [n]} (f(x_i; w_1, w_2) - y_i) \sigma(w_1 x_i) - \sigma(w_1 x_i) \zeta_{i,t}.$$

This can be rewritten as a pair of SDEs.

$$dw_{1,t} = -\frac{1}{n} \sum_{i \in [n]} (f(x_i; w_1, w_2) - y_i) w_2 x_i \sigma'(w_1 x_i) dt + \frac{\sqrt{\eta}\zeta}{\sqrt{n}} \sum_{i \in [n]} w_2 x_i \sigma'(w_1 x_i) dB_{i,t}$$

$$dw_{2,t} = -\frac{1}{n} \sum_{i \in [n]} (f(x_i; w_1, w_2) - y_i) \sigma(w_1 x_i) dt + \frac{\sqrt{\eta}\zeta}{\sqrt{n}} \sum_{i \in [n]} \sigma(w_1 x_i) dB_{i,t}, \quad (31)$$

where $\zeta > 0$ is a constant representing the amplitude of the gradient noise, $\eta > 0$ is the learning rate, and $B_{i,t}$ denotes the $n$-dimensional standard Wiener process. Note that the rescaling factor $1/\sqrt{n}$ in the diffusion term is introduced to avoid the data size $n$ influencing the SDE's behavior.

## F.1 Proof of Theorem 5.1

As a preparation, we first show a modified version of Theorem E.1 to incorporate stopped processes.

**Lemma F.1.** *Let $A \subset \mathbb{R}^d$ denote an affine subset and $A_c$ be an open neighborhood around $A$ defined as $A_c = \{\theta \in \mathbb{R}^d : \inf_{x \in A} \|\theta - x\| < c\}$, where $c > 0$ is a constant. Consider a stochastic process $\{\theta_t \in \mathbb{R}^d : t \geq 0\}$ that follows the SDE Eq. 25 within $A_c$ with all the regularity conditions mentioned in Theorem E.1. Futhermore, let $\tau = \inf\{t \geq 0 : \theta_t \notin A_c \cap \text{int } B\}$, where $B = \{\theta \in \mathbb{R}^d : \|\theta_i\| \leq c_B\}$ with $i \in [d]$ and $c_B > 0$. If $A \cap B$ is an invariant set of $\theta_t$, then $A \cap B$ is a stochastically attractive invariant set of the stopped process $\{\theta_{t \wedge \tau} : t \geq 0\}$, provided Eq. 26 and and 27 hold for any $x \in A \cap B$.*

*Proof.* Since the drift and diffusion terms in Eq.25 are Lipschitz continuous within $A_c$, there exists a unique solution for the SDE up to the exit time $\tau$. This allows us to consider the unique stopped process $\theta_{t \wedge \tau}$, a continuous strongly Markov process. Let $l(\theta) : \mathbb{R}^d \to \mathbb{R}_{\geq 0}$ be the Euclidian distance between $\theta \in \mathbb{R}^d$ and the invariant set $A \cap B$, and $\tilde{\mathcal{A}}$ be the infinitesimal operator of $\theta_{t \wedge \tau}$ where $\tau_\epsilon = \inf\{t \geq 0 : \theta_t \notin A_\epsilon\}$. The same argument in Theorem E.1 leads to $\tilde{\mathcal{A}}(l^\lambda(x)) \leq 0$ for any $x \in A_\epsilon \cap B$ with small enough $\epsilon > 0$ and $\lambda > 0$. By applying a local version of maximial inequality (Lemma 1 in Kushner [44]), we obtain, for any $x \in A_\epsilon$ and $0 < \epsilon' < \epsilon$,

$$\mathbb{P}\left[\sup_{t \geq 0} l(\theta_{t \wedge \tau}) \geq \epsilon' \,\middle|\, \theta_0 = x\right] = \mathbb{P}\left[\sup_{t \geq 0} l(\theta_{t \wedge \tau \wedge \tau_\epsilon}) \geq \epsilon' \,\middle|\, \theta_0 = x\right] \leq \frac{l^\lambda(x)}{\epsilon'}.$$

This implies that $A \cap B$ is a stochastically attractive invariant set of the stopped process $\theta_{t \wedge \tau}$. $\square$

**Theorem 5.1** (formal). *Suppose $\{(w_{1,t}, w_{2,t}) \in \mathbb{R}^2 : t \geq 0\}$ is a stochastic process obeying Eq. 31 in a neighborhood $U$ of $A = \{(0,0)\} \subset \mathbb{R}^2$, and the activation function $\sigma : \mathbb{R} \to \mathbb{R}$ is a smooth function such that $\sigma(0) = 0$. The invariant set $A$ is stochastically attractive if $\frac{\eta\zeta^2 |\sigma'(0)|}{2} > \frac{|\sum_{i \in [n]} x_i y_i|}{\sum_{i \in [n]} x_i^2}$.*

*Proof.* If $\sigma'(0) = 0$, the condition can never be met. Therefore, we only consider the case $\sigma'(0) \neq 0$. Without loss of generality, assume that $U$ is an open ball centered at $(0,0)$. Since the drift and diffusion terms of Eq.31 are locally Lipschitz continuous, it has a unique solution up to the exit time of $U$. We define stochastic processes $p_t = w_{t,1} + w_{t,2}$, $s = w_{t,1} - w_{t,2}$. These two processes obey the following SDEs:

$$\begin{cases} dp_t = \mu_p(p_t, s_t)dt + \frac{1}{\sqrt{n}} \sum_{i \in [n]} \sigma_{p,i}(p_t, s_t)dB_{i,t} \\ ds_t = \mu_s(p_t, s_t)dt + \frac{1}{\sqrt{n}} \sum_{i \in [n]} \sigma_{s,i}(p_t, s_t)dB_{i,t} \end{cases}$$

where

$$\begin{cases} \mu_p(p,s) = -\frac{1}{n} \sum_{i \in [n]} [(f(x_i; w_{1,t}, w_{2,t}) - y_i)w_{2,t}\sigma'(w_{1,t}x_i)x_i + (f(x_i; w_{1,t}, w_{2,t}) - y_i)\sigma(w_{1,t}x_i)] \\ \mu_s(p,s) = -\frac{1}{n} \sum_{i \in [n]} [(f(x_i; w_{1,t}, w_{2,t}) - y_i)w_{2,t}\sigma'(w_{1,t}x_i)x_i - (f(x_i; w_{1,t}, w_{2,t}) - y_i)\sigma(w_{1,t}x_i)] \\ \sigma_{p,i}(p,s) = \sqrt{\eta}\zeta w_{2,t}\sigma'(w_{1,t}x_i)x_i + \sqrt{\eta}\zeta\sigma(w_{1,t}x_i) \\ \sigma_{s,i}(p,s) = \sqrt{\eta}\zeta w_{2,t}\sigma'(w_{1,t}x_i)x_i - \sqrt{\eta}\zeta\sigma(w_{1,t}x_i) \end{cases}$$

Define stopping times $\tau_p = \inf_{t \geq 0}\{p_t \geq \epsilon\}$, $\tau_s = \inf_{t \geq 0}\{s_t \geq \epsilon\}$ where $\epsilon > 0$ satisfies $\{(p,s) : \max(|p|, |s|) < \epsilon\} \subset U$, and consider the stopped processes $(p_{t \wedge \tau_p}, s_{t \wedge \tau_p})$ and $(p_{t \wedge \tau_s}, s_{t \wedge \tau_s})$. We have,

$$\frac{\partial}{\partial p}\mu_p(0,0) - \frac{1}{4n}\frac{\partial^2}{\partial p^2} \sum_{i \in [n]} \sigma_{p,i}^2(0,0) \quad = \quad -\frac{\sigma'(0)}{n} \sum_{i \in [n]} y_i x_i - \frac{1}{2n}\eta\zeta^2\sigma'(0)^2 \sum_{i \in [n]} x_i^2 < 0$$

$$\frac{1}{n}\frac{\partial^2}{\partial p^2} \sum_{i \in [n]} \sigma_{p,i}^2(0,0) \quad = \quad \frac{1}{2n}\eta\zeta^2\sigma'(0)^2 \sum_{i \in [n]} x_i^2 > 0,$$

By the assumption on smoothness, $\frac{\partial}{\partial p}\mu_p(0,s) - \frac{1}{4}\frac{\partial^2}{\partial p^2} \sum_{i \in [n]} \sigma_{p,i}^2(0,s) < 0$ and $\frac{\partial^2}{\partial p^2} \sum_{i \in [n]} \sigma_{p,i}^2(0,s) > 0$ for any $s \in [-\epsilon, \epsilon]$ with small enough $\epsilon > 0$. Therefore by Lemma F.1, $A_p := \{(0,s) : s \in [-\epsilon, \epsilon]\}$ is stochastically attractive for the process $\{(p_{t \wedge \tau_p}, s_{t \wedge \tau_p}) : t \geq 0\}$. Similarly,

$$\frac{\partial}{\partial s}\mu_s(0,0) - \frac{1}{4n}\frac{\partial^2}{\partial s^2} \sum_{i \in [n]} \sigma_{s,i}^2(0,0) \quad = \quad \frac{\sigma'(0)}{n} \sum_{i \in [n]} y_i x_i - \frac{1}{2n}\eta\zeta^2\sigma'(0)^2 \sum_{i \in [n]} x_i^2 < 0$$

$$\frac{1}{n}\frac{\partial^2}{\partial s^2} \sum_{i \in [n]} \sigma_{s,i}^2(0,0) \quad = \quad \frac{1}{2n}\eta\zeta^2\sigma'(0)^2 \sum_{i \in [n]} x_i^2 > 0.$$

By Lemma F.1, $A_s := \{(p,0) : p \in [-\epsilon, \epsilon]\}$ is stochastically attractive for the process $\{(p_{t \wedge \tau_s}, s_{t \wedge \tau_s}) : t \geq 0\}$.

By combining these two facts, we see that $A_p \cap A_s = \{(0,0)\}$ is stochastically attractive for the process $\{(w_{1,t}, w_{2,t}) : t \geq 0\}$. $\qquad \square$

# G The permutation invariant set in one hidden layer nonlinear neural network

Consider training a two neuron neural network $f_\theta$ on a dataset $\{(x_k, y_k)\}_{k=1}^N$, where $f_\theta(x) = \sum_{i=1}^2 w_i^{(2)} \sigma(w_i^{(1)} x)$. We assume the data are drawn randomly from standard Gaussian distribution. Consider running gradient descent with learning rate $\eta$ on the mean square error loss with label noise drawn freshly from $\mathcal{N}(0, \sigma^2)$. In this set up, the invariant set from the permutation symmetry is the affine space $\{\theta | w_1^{(i)} = w_2^{(i)}$ for $i = 1, 2\}$. Empirically, we observe the stochastic collapse towards this invariant set as shown in 7. We found that the speed of collapsing depends on both the learning rate and the noise scale. Interestingly, the attractivity strengthens with increased learning rate and noise scale up to a certain threshold.

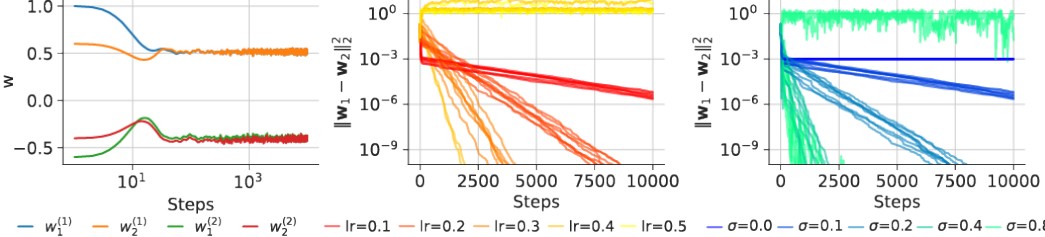

Figure 7: **Gradient noises in SGD induces stochastic collapsing to the invariant set associated with identical neurons**. **Left**: The collapsing process of the weights of hidden neurons 1 and 2 becoming identical over the course of training, demonstrating SGD's bias towards simpler subnetworks. **Middle**: A visualization of the impact of learning rates on the stochastic attractivity. An increase in the learning rate accelerates the collapsing process until reaching a critical threshold, beyond which the collapsing ceases. **Right**: The influence of noise scale on the collapsing process. The attractivity strengthens with increased noise up to a certain threshold.

# H Conditions of sign and permutation invariant sets in the presence of residual connection

**Definition H.1** (Residual connection). *A hidden layer $l_2$ in a neural network is defined as residually connected from a previous layer $l_1$ if its forward pass follows:*

$$h^{(l_2)}(x) = \sigma \left( w^{(l_2-1)} h^{(l_2-1)}(x) + b^{(l_2)} \right) + h^{(l_1)}(x),$$

*where $h^{(l)}$ denotes the hidden activations of layer $l$, $w^{(l)}$ is the weight matrix at layer $l$, and $b^{(l)}$ is the bias vector at layer $l$.*

**Proposition H.1** (Sign invariant set for residual neural network). *Consider a hidden neuron $p$ of layer $l_2$ with a residual connection from layer $l_1$. Let $(w_{in,p}^{(l)}, b_p^{(l)})$ denote the parameters (weights and bias) directly incoming to this neuron, and $w_{out,p}^{(l+1)}$ represent the parameters (weights) directly outgoing from the neuron. If the non-linearity $\sigma$ is origin-passing (i.e. $\sigma(0) = 0$), then the coordinate plane $A = \{\theta | w_{in,p}^{(l)} = 0, b_p^{(l)} = 0, w_{out,p}^{(l+1)} = 0 \text{ for } l = l_1, l_2\}$ forms an invariant set.*

**Proposition H.2** (Permutation invariant set for residual neural network). *Consider two hidden neurons $p, q$ in the same layer $l_1$ with a residual connection from layer $l_1$. Let $(w_{in,p}^{(l)}, b_p^{(l)}), (w_{in,q}^{(l)}, b_q^{(l)})$ denote the parameters directly incoming to the neurons, and $w_{out,p}^{(l+1)}, w_{out,q}^{(l+1)}$ the parameters directly outgoing from the neurons. The affine space $A = \{\theta | w_{in,p}^{(l)} = w_{in,q}^{(l)}, b_p^{(l)} = b_q^{(l)}, w_{out,p}^{(l+1)} = w_{out,q}^{(l+1)} \text{ for } l = l_1, l_2\}$ is an invariant set.*

We omit the proofs as they closely resemble the proofs in App. B.1 and B.1. In the presence of residual connections, the conditions for the sign and permutation invariant sets become stronger. Not only do the weights and biases of the neuron(s) at layer $l_2$ need to follow the condition, the neuron(s) which are residually connected from must also must also meet these conditions.

# I Understanding Stochastic Collapse within a Teacher-Student Framework

To investigate the impact of the stochastic collapse on generalization, we will extend the gradient flow analysis of training error [53] and test error [51] for two-layer linear neural networks to stochastic gradient flow.

## I.1 The Linear Teacher-Student Framework

We follow the teacher-student framework proposed by Lampinen and Ganguli [51].

**Low-rank teacher network.** We consider a low-rank teacher that computes the linear map $\bar{W}$ : $\mathbb{R}^n \to \mathbb{R}^m$. $\bar{W} \in \mathbb{R}^{m \times n}$ is a low-rank matrix where $k \ll \min\{m, n\}$ denotes the rank.

**Two-layer student network.** We consider a two-layer student network, parameterized by $W_1 \in \mathbb{R}^{d \times n}$ and $W_2 \in \mathbb{R}^{m \times d}$, that computes the linear map $\hat{W} = W_2 W_1 : \mathbb{R}^n \to \mathbb{R}^m$. Here $d$ is the hidden-layer dimension and we let $d = \min\{m, n\}$ such that the student has the capacity to represent all linear maps from $\mathbb{R}^n$ to $\mathbb{R}^m$.

**Noisy training data.** The student network is trained on data generated by the teacher network through the noisy map,

$$y = \bar{y} + \epsilon \quad \text{where} \quad \bar{y} = \bar{W}x, \tag{32}$$

and $\epsilon \sim \mathcal{N}(0, \sigma_\epsilon^2 I_m)$. We will consider a fixed training dataset of $p$ noisy input-output pairs $\{(x_1, y_1), \ldots, (x_p, y_p)\}$ generated by the teacher network from input $x_i \sim \mathcal{N}(0, \sigma_x^2 I_n)$. Let $X \in \mathbb{R}^{n \times p}$ and $Y \in \mathbb{R}^{m \times p}$ be the matrices with columns $x_i$ and $y_i$ respectively. This setup yields the second-order training statistics that guide the student network's learning dynamics,

$$\Sigma_{xx} = XX^\intercal = \sum_{i=1}^p x_i x_i^\intercal, \qquad \Sigma_{yx} = YX^\intercal = \sum_{i=1}^p y_i x_i^\intercal. \tag{33}$$

Later in our analysis of the learning dynamics we will assume the input data matrix $X$ is whitened such that $\Sigma_{xx} = I_n$ implying that only the input-output covariance structure $\Sigma_{yx}$ will govern the learning dynamics of the student.

**Train and test error.** The student network is trained to minimize the mean squared error loss between the prediction $\hat{Y} = \hat{W}X$ and the output $Y$, and evaluated on the expected prediction error for a new test point sampled from the input distribution. We denote the train and test error as

$$\mathcal{E}_{\text{train}} = \frac{\|\hat{W}X - Y\|_F^2}{\|Y\|_F^2}, \qquad \mathcal{E}_{\text{test}} = \frac{\mathbb{E}\left[\|\hat{W}x - \bar{W}x\|_F^2\right]}{\mathbb{E}\left[\|\bar{W}x\|_F^2\right]}, \tag{34}$$

where $x \sim \mathcal{N}(0, \sigma_x^2 I_n)$ is a new test point sampled from the input distribution, $\|\cdot\|_F$ denotes the Frobenius norm, and $\mathbb{E}[\cdot]$ represents the expectation over the training data and the input distribution of the test point.

**Singular value structure.** The train and test performance of a student is determined by the relationship between three matrices in $\mathbb{R}^{m \times n}$: the low rank teacher $\bar{W}$, the overparameterized student $\hat{W}$, and the noisy training data $\Sigma_{yx}$. Due to the linear nature of this problem, we will use the Singular Value Decomposition (SVD) for all three matrices to describe their relationship,

$$\bar{W} = \sum_{i=1}^k \bar{s}_i \bar{u}_i \bar{v}_i^\intercal, \qquad \hat{W} = \sum_{i=1}^d \hat{s}_i \hat{u}_i \hat{v}_i^\intercal, \qquad \Sigma_{yx} = \sum_{i=1}^d \tilde{s}_i \tilde{u}_i \tilde{v}_i^\intercal. \tag{35}$$

Here, the $s_i$ denote non-zero singular values, and $u_i$ and $v_i$ denote the left and right singular vectors for their respective matrices. We will sometimes also find it useful to concatenate this information into a matrix representation where we will use $U$, $S$, and $V$ with the appropriate accents $\bar{\cdot}$, $\hat{\cdot}$, or $\tilde{\cdot}$ to denote which matrix they are associated with.

## I.2 Exact Solutions to the Student Learning Dynamics

The student network is trained by gradient descent with learning rate $\eta$ to minimize $\mathcal{E}_{\text{train}}$. However, we assume that each gradient evaluation is corrupted with Gaussian label-noise such that the training

dynamics are given by the coupled update equations,

$$W_1^{(t+1)} = W_1^{(t)} - \eta \left( W_2^{(t)} \right)^\intercal \left( W_2^{(t)} W_1^{(t)} \Sigma_{xx} - \Sigma_{yx} + Z^{(t)} X^\intercal \right), \tag{36}$$

$$W_2^{(t+1)} = W_2^{(t)} - \eta \left( W_2^{(t)} W_1^{(t)} \Sigma_{xx} - \Sigma_{yx} + Z^{(t)} X^\intercal \right) \left( W_1^{(t)} \right)^\intercal, \tag{37}$$

where $W_1^{(t)}$ and $W_2^{(t)}$ are the parameters after $t$ steps of training and $Z^{(t)} \in \mathbb{R}^{m \times p}$ is a matrix of label-noise associated with step $t$. Directly studying these dynamics is difficult because the gradients equations are coupled between the weights $W_1$ and $W_2$. However, by introducing some assumptions on the covariance of the inputs and the label-noise, selecting a special initialization of the parameters, and taking the limit as $\eta \to 0$, we can decouple the dynamics into a system of scalar non-linear SDEs with exact solutions.

**Decoupling the dynamics.** We introduce the following assumptions:

A1. *Whitened-Input.* We assume that $\Sigma_{xx} = I_n$ such that the input data matrix $X$ is whitened (this implicitly assumes $p \geq n$ such that we have at least as many observations as features).

A2. *Structured Label-Noise.* We assume the the gradient noise can be decomposed as $Z^{(t)} = \tilde{U} \mathrm{diag}\left( z^{(t)} \right) \tilde{V}^\intercal X$ where $z^{(t)} \sim \mathcal{N}(0, \zeta^2)$.

A3. *Spectral Initialization.* We assume that the student network is initialized such that $W_1^{(0)} = O A^{(0)} \tilde{V}^\intercal$ and $W_2^{(0)} = \tilde{U} B^{(0)} O^\intercal$ where $A^{(0)}$ and $B^{(0)}$ are diagonal matrices such that $\hat{S}(0) = B^{(0)} A^{(0)}$ and $O \in \mathbb{R}^{d \times d}$ is a random orthonormal matrix such that $O^\intercal O = I_d$.

A4. *Balanced Initialization.* We assume a balanced initialization such that $W_1^{(0)} W_1^{(0)\intercal} = W_2^{(0)\intercal} W_2^{(0)}$.

Given the first three assumptions, the dynamics decouple in the eigenbasis of $\Sigma_{yx}$. Under the change of variables $A^{(t)} = O^\intercal W_1^{(t)} \tilde{V}$ and $B^{(t)} = \tilde{U}^\intercal W_2^{(t)} O$, the dynamics transform to

$$A^{(t+1)} = A^{(t)} - \eta \left( B^{(t)} \right)^\intercal \left( B^{(t)} A^{(t)} - \tilde{S} + \mathrm{diag}\left( z^{(t)} \right) \right), \tag{38}$$

$$B^{(t+1)} = B^{(t)} - \eta \left( B^{(t)} A^{(t)} - \tilde{S} + \mathrm{diag}\left( z^{(t)} \right) \right) \left( A^{(t)} \right)^\intercal, \tag{39}$$

which because $A^{(0)}$ and $B^{(0)}$ are diagonal matrices by assumption, decouples into a system of scalar equations. Taking the limit as $\eta \to 0$ we can approximate each of these scalar equations with a two-dimensional non-linear SDE,

$$d \begin{bmatrix} a_i \\ b_i \end{bmatrix} = - \begin{bmatrix} b_i \left( a_i b_i - \tilde{s}_i \right) \\ a_i \left( a_i b_i - \tilde{s}_i \right) \end{bmatrix} dt + \sqrt{\eta \zeta^2} \begin{bmatrix} b_i \\ a_i \end{bmatrix} dB_t, \tag{40}$$

where $a_i$ and $b_i$ are the $i^{th}$ diagonal element of $A$ and $B$ respectively. Under these much simpler dynamics, it is not difficult to prove by Itô's Lemma that the fourth assumption will be obtained no matter the initialization,

**Lemma I.1** (Autobalancing). *Under the dynamics in equation (40), the $\lim_{t \to \infty} a_i(t)^2 - b_i(t)^2 = 0$.*

*Proof.* Let $r_i(t)$ denote the difference $r_i(t) = a_i(t)^2 - b_i(t)^2$. By Itô's Lemma, $r_i(t)$ is driven by the ODE $dr_i = -\eta \zeta^2 r_i dt$, which has the temporal solution $r_i(t) = r_i(0) e^{-\eta \zeta^2 t}$. Because $\eta \zeta^2 > 0$, then the $\lim_{t \to \infty} r_i(t) = 0$. $\qquad \square$

The result of all four assumptions A1 - A4 is a decoupling the learning dynamics into a set of independent quartic slices of the loss landscape, $\ell_i(w_i) = (w_i^2 - \tilde{s}_i)^2$ with $w_i = a_i = b_i$, where within each slice the dynamics evolve according to the non-linear SDE,

$$dw_i = -w_i(w_i^2 - \tilde{s}_i)dt + \sqrt{\eta \zeta^2} w_i dB_t. \tag{41}$$

The geometry of these slices is controlled by the strength of the singular mode for the training data $\tilde{s}_i$ and the dynamics are determined by the relationship with the student's singular mode strength $\hat{s}_i = w_i^2$. By Itô's Lemma we find that the dynamics of $\hat{s}_i$ are governed by the non-linear SDE,

$$d\hat{s}_i = 2\hat{s}_i \left( \left( \tilde{s}_i + \frac{\eta \zeta^2}{2} \right) - \hat{s}_i \right) dt + 2\sqrt{\eta \zeta^2} \hat{s}_i dB_t, \tag{42}$$

which is an equation often used to model population growth in a stochastic, crowded environment. See Oksendal [58], Chapter 5 problem 5.15 for more details.

**Theorem 6.1.** *Under assumptions A1 - A4, the dynamics of $\hat{s}_i(t)$ for $t \geq 0$ are governed by the stochastic process,*

$$\hat{s}_i(t) = \frac{e^{(2\tilde{s}_i - \eta\zeta^2)t + 2\sqrt{\eta\zeta^2}B_t}}{2\int_0^t e^{(2\tilde{s}_i - \eta\zeta^2)\tau + 2\sqrt{\eta\zeta^2}B_\tau}d\tau + \hat{s}_i(0)^{-1}}. \tag{43}$$

*Proof.* Let $u_i = \hat{s}_i^{-1}$. Then by Itô's Lemma, the dynamics of $u_i$ are given by the SDE,

$$du_i = \left(2 - (2\tilde{s}_i - 3\eta\zeta^2)u_i\right)dt - 2\sqrt{\eta\zeta^2}u_i dB_t \tag{44}$$

Rearranging this SDE we can express it in the standard form of a geometric Ornstein–Uhlenbeck process (sometimes referred to as a modified Ornstein-Uhlenbeck process)

$$du_i = \theta(\mu - u_i)dt + \sigma u_i dB_t \tag{45}$$

where $\theta = 2\tilde{s}_i - 3\eta\zeta^2$, $\mu = (\tilde{s}_i - \frac{3}{2}\eta\zeta^2)^{-1}$, and $\sigma = -2\sqrt{\eta\zeta^2}$. To derive a solution to this linear SDE we make the ansatz that the solution takes the form $u_i(t) = F(t)G(t)$ where $F(t)$ and $G(t)$ solve the respective SDEs,

$$dF = -\theta F dt + \sigma F dB_t, \qquad dG = \alpha(t)dt + \beta(t)dB_t. \tag{46}$$

To determine the unknown coefficients $\alpha(t)$ and $\beta(t)$ defining the SDE for $G(t)$, we can apply Itô's product rule

$$\begin{aligned}
du_i &= dFG + FdG + dFdG \\
&= (-\theta FG + \alpha(t)F + \beta(t)\sigma F)dt + (\sigma FG + \beta(t)F)dB_t \\
&= (\alpha(t)F + \beta(t)\sigma F - \theta u_i)dt + (\sigma u_i + \beta(t)F)dB_t
\end{aligned}$$

Aligning this expression with the original SDE for $u_i$ we see that

$$\alpha(t) = \theta\mu F^{-1}, \qquad \beta(t) = 0. \tag{47}$$

We can now solve for $F(t)$ and $G(t)$. The SDE defining $F(t)$ is standard geometric Brownian motion and thus has the solution,

$$F(t) = e^{-\left(\theta + \frac{1}{2}\sigma^2\right)t + \sigma B_t} \tag{48}$$

where we assumed $F(0) = 1$ and thus $G(0) = u_i(0)$. Using this solution for $F(t)$, then $G(t)$ is given by

$$G(t) = u_0 + \theta\mu\int_0^t e^{\left(\theta + \frac{1}{2}\sigma^2\right)\tau - \sigma B_s}d\tau. \tag{49}$$

Combining our solutions for $F(t)$ and $G(t)$ we get that the solution for $u_i(t)$ is

$$u_i(t) = u_i(0)e^{-\left(\theta + \frac{1}{2}\sigma^2\right)t + \sigma B_t} + \theta\mu\int_0^t e^{-\left(\theta + \frac{1}{2}\sigma^2\right)(t-\tau) + \sigma(B_t - B_\tau)}d\tau. \tag{50}$$

Inverting this expression and rearranging terms, gives the expression for $\hat{s}_i(t)$,

$$\hat{s}_i(t) = u_i(t)^{-1} = \frac{e^{\left(\theta + \frac{1}{2}\sigma^2\right)t - \sigma B_t}}{\theta\mu\int_0^t e^{\left(\theta + \frac{1}{2}\sigma^2\right)\tau - \sigma B_\tau}d\tau + \hat{s}_i(0)^{-1}} \tag{51}$$

Plugging in the simplifications $\theta\mu = 2$, $\theta + \frac{1}{2}\sigma^2 = 2\tilde{s}_i - \eta\zeta^2$, and $\sigma = -2\sqrt{\eta\zeta^2}$ gives the final expression. $\qquad\square$

**Corollary 6.1.** *In the limit of $\hat{s}_i(0) \to 0$, for any $t > 0$, $\hat{s}_i(t)$ is given by*

$$\hat{s}_i(t)/\hat{s}_i(0) \underset{a.s.}{\to} e^{(2\tilde{s}_i - \eta\zeta^2)t + 2\sqrt{\eta\zeta^2}B_t}. \tag{52}$$

*Proof.* Since $B_t$ is almost surely continuous, $(2\tilde{s}_i - \eta\zeta^2)\tau + 2\sqrt{\eta\zeta^2}B_\tau$ has a maximum in $\tau \in [0, t]$ and therefore $2\int_0^t e^{(2\tilde{s}_i - \eta\zeta^2)\tau + 2\sqrt{\eta\zeta^2}B_\tau}\,d\tau$ is finite almost surely. Therefore, by Theorem 6.1,

$$\hat{s}_i(t)/\hat{s}_i(0) = \frac{e^{(2\tilde{s}_i - \eta\zeta^2)t + 2\sqrt{\eta\zeta^2}B_t}}{2\hat{s}_i(0)\int_0^t e^{(2\tilde{s}_i - \eta\zeta^2)\tau + 2\sqrt{\eta\zeta^2}B_\tau}\,d\tau + 1} \underset{a.s.}{\to} e^{(2\tilde{s}_i - \eta\zeta^2)t + 2\sqrt{\eta\zeta^2}B_t}. \tag{53}$$

$\square$

Corollary 6.1 has important implications for the dynamics of $\hat{s}_i(t)$. It demonstrates that the dynamics observe a phase transition determined by the ratio $2\tilde{s}_i/\eta\zeta^2$, consistent with Theorem 4.1. When this ratio is less than one, the distribution will exhibit stochastic collapse, converging to a delta distribution at the origin. When this ratio is greater than one, the distribution will converge to a stationary distribution with a positive expectation. Corollary 6.1 demonstrates that stochastic collapse is determined by a signal-to-noise ratio where the signal derives from the noisy teacher and the noise originates from stochastic gradients. While Corollary 6.1 describes the dynamics with vanishing initialization for any $t > 0$, it does not describe what happens with finite initialization nor infinite time. To understand these scenarios we consider the stationary solution for the dynamics of $\hat{s}_i$.

**Corollary I.1** (Stationary solution). *In the limit of $t \to \infty$, $\hat{s}_i(t)$ is distributed according to the density,*

$$p_{ss}(\hat{s}_i) = Z^{-1}e^{-\frac{\hat{s}_i}{\eta\zeta^2}}\hat{s}_i^{-\left(\frac{3}{2} - \frac{\tilde{s}_i}{\eta\zeta^2}\right)} \tag{54}$$

*where $Z = \int_0^{+\infty} e^{-\frac{\hat{s}_i}{\eta\zeta^2}}\hat{s}_i^{-\left(\frac{3}{2} - \frac{\tilde{s}_i}{\eta\zeta^2}\right)}\,d\hat{s}_i$ is the partition function.*

*Proof.* To check if $p_{ss}(\hat{s}_i)$ is the stationary solution we use the *Fokker-Planck equation* for the dynamics of $\hat{s}_i$, which is

$$\partial_t p = \nabla \cdot \underbrace{\left(2\hat{s}_i\left(\hat{s}_i - \left(\tilde{s}_i + \frac{\eta\zeta^2}{2}\right)\right)p + 2\eta\zeta^2\nabla \cdot \left(\hat{s}_i^2 p\right)\right)}_{-J}, \qquad p(\hat{s}_i, 0) = \delta_{\hat{s}_i(0)}, \tag{55}$$

where $J$ is commonly referred to as the *probability current*. The *stationary solution* $p_{ss}$ satisfies $\partial_t p_{ss} = 0$, which because we are considering a one-dimensional SDE, is equivalent to $J_{ss} = 0$. Plugging in the expression for $p_{ss}(\hat{s}_i)$ and using the simplification

$$2\eta\zeta^2\nabla \cdot \left(\hat{s}_i^2 p\right) = 4\zeta^2\hat{s}_i p_{ss} + 2\eta\zeta^2\hat{s}_i^2\nabla p_{ss}(\hat{s}_i) \tag{56}$$

$$= 4\zeta^2\hat{s}_i p_{ss} - 2\eta\zeta^2\hat{s}_i^2\left((\eta\zeta^2)^{-1} + \hat{s}_i^{-1}\left(\frac{3}{2} - \frac{\tilde{s}_i}{\eta\zeta^2}\right)\right)p_{ss}(\hat{s}_i) \tag{57}$$

$$= -2\hat{s}_i\left(\hat{s}_i - \left(\tilde{s}_i + \frac{\eta\zeta^2}{2}\right)\right)p_{ss} \tag{58}$$

we can verify that $J_{ss} = 0$. $\square$

The stationary solution can be interpreted as a Gibb's distribution (i.e. $p_{ss} \propto e^{-\kappa\Psi(x)}$) with the potential

$$\Psi(\hat{s}_i) = \hat{s}_i + \left(\frac{3\eta\zeta^2}{2} - \tilde{s}_i\right)\log(\hat{s}_i) \tag{59}$$

and temperature constant $\kappa = \left(\eta\zeta^2\right)^{-1}$. From this interpretation we see two phase transitions,

$$\arg\max p_{ss}(\hat{s}_i) = \begin{cases} \tilde{s}_i - \frac{3}{2}\eta\zeta^2 & \text{if } \tilde{s}_i > \frac{3}{2}\eta\zeta^2 \\ 0 & \text{if } \tilde{s}_i \leq \frac{3}{2}\eta\zeta^2 \end{cases} \qquad p_{ss}(\hat{s}_i) = \begin{cases} Z^{-1}e^{-\kappa\Psi(w)} & \text{if } \tilde{s}_i > \frac{\eta\zeta^2}{2} \\ \delta_0 & \text{if } \tilde{s}_i \leq \frac{\eta\zeta^2}{2} \end{cases} \tag{60}$$

When $\frac{3}{2}\eta\zeta^2 \leq \tilde{s}_i$, the most probable state is $\tilde{s}_i - \frac{3}{2}\eta\zeta^2$, where we already see a bias from the stochasticity towards the origin. When $\frac{3}{2}\eta\zeta^2 > \tilde{s}_i$, the most probable state is at the origin. When $\frac{\eta\zeta^2}{2} \geq \tilde{s}_i$, the partition function $Z$ diverges and the stationary distribution collapses to a delta distribution at the origin, which is an invariant set.

We now conjecture an exact expression for the temporal dynamics of the expectation $\mathbb{E}[\hat{s}_i(t)]$:

**Conjecture I.1.** *Under assumptions A1 - A4, the expectation $\mathbb{E}[\hat{s}_i(t)]$ for $t \geq 0$ is given by the equation,*

$$\mathbb{E}[\hat{s}_i(t)] = \frac{(\tilde{s}_i - \frac{\eta\zeta^2}{2})e^{(2\tilde{s}_i - \eta\zeta^2)t}}{e^{(2\tilde{s}_i - \eta\zeta^2)t} - 1 + \frac{\tilde{s}_i - \frac{\eta\zeta^2}{2}}{\hat{s}_i(0)}}. \tag{61}$$

Conjecture I.1 captures both the phase transition for stochastic collapse from Corollary 6.1 and the student's singular value shrinkage from Corollary I.1. Furthermore, in the limit of $\zeta \to 0$, it aligns with the dynamics from Theorem 6.1, as would be expected. Additionally, one can show that the dynamics predicted by Conjecture I.1 are equivalent to the deterministic dynamics given by $L_2$ regularized gradient flow with a regularization coefficient of $\lambda = \frac{\eta\zeta^2}{2}$.

## I.3 An Analytic Theory of Error Dynamics in the High-dimensional Limit

So far we have setup a teacher-student setting and demonstrated that the learning dynamics of the student can be decomposed into...nonlinear... Now will show how this implicit shrinking and collapsing process is an effective form of regularization that improves the performance of the student

**Decomposition of the error.** Using the the SVD structure of the low-rank teacher $\{\bar{u}_i, \bar{s}_i, \bar{v}_i\}$, overparameterized student $\{\hat{u}_i, \hat{s}_i, \hat{v}_i\}$, and noisy training data $\{\tilde{u}_i, \tilde{s}_i, \tilde{v}_i\}$ we can decompose these error terms as

$$\mathcal{E}_{\text{train}} = \left(\sum_{i=1}^{m} \tilde{s}_i^2\right)^{-1} \left(\sum_{i=1}^{m} \tilde{s}_i^2 + \sum_{j=1}^{d} \hat{s}_j^2 - 2\sum_{i=1}^{m}\sum_{j=1}^{d} \tilde{s}_i \hat{s}_j \langle \tilde{u}_i, \hat{u}_j\rangle\langle\tilde{v}_i, \hat{v}_j\rangle\right), \tag{62}$$

$$\mathcal{E}_{\text{test}} = \left(\sum_{i=1}^{k} \bar{s}_i^2\right)^{-1} \left(\sum_{i=1}^{k} \bar{s}_i^2 + \sum_{j=1}^{d} \hat{s}_j^2 - 2\sum_{i=1}^{k}\sum_{j=1}^{d} \bar{s}_i \hat{s}_j \langle \bar{u}_i, \hat{u}_j\rangle\langle\bar{v}_i, \hat{v}_j\rangle\right). \tag{63}$$

From these expressions we see that understanding the dynamics of the train and test error depends on (1) the alignment of the singular values and vectors of the student network with the SVD structure of the training data and (2) the relationship between the SVD structure of the training data and that of the teacher network. Our analysis of the student network training dynamics covers (1) and as done in Lampinen and Ganguli [51] we can understand (2) in the high-dimensional limit through a random matrix theory analysis.

**Random matrix theory analysis.** We will work in the limit $n, m \to \infty$ with a finite rank $k \sim O(1)$ and finite aspect ratio $\mathcal{A} = \frac{m}{n} \in (0, \infty)$. Without loss of generality, we assume $\sigma_\epsilon^2 = n^{-1}$ such that the singular values of the noise are $O(1)$ and thus the singular values of the teacher network can be understood as signal to noise ratios (SNRs). The relationship between the SVD structures of a low-rank matrix (the teacher network $\{\bar{u}_i, \bar{s}_i, \bar{v}_i\}$) and a noisy perturbation (the training data $\{\tilde{u}_i, \tilde{s}_i, \tilde{v}_i\}$) is well studied in [59], which we summarize here.

In the limit, the top $k$ singular values of $\Sigma_{yx}$ converge to values given by the transfer function,

$$\tilde{s}(\bar{s}) = \begin{cases} (\bar{s})^{-1}\sqrt{(1 + \bar{s}^2)(\mathcal{A} + \bar{s}^2)} & \text{if } \bar{s} > \mathcal{A}^{1/4} \\ 1 + \mathcal{A} & \text{otherwise} \end{cases} \tag{64}$$

while the bottom $m - k$ singular values are distributed according to the Marchenko-Pastur (MP) distribution,

$$p_{MP}(\tilde{s}) = \begin{cases} \frac{\sqrt{4\mathcal{A} - (\tilde{s}^2 - (1+\mathcal{A}))^2}}{\pi \mathcal{A}\tilde{s}} & \text{if } \tilde{s} \in [1 - \sqrt{\mathcal{A}}, 1 + \sqrt{\mathcal{A}}] \\ 0 & \text{otherwise} \end{cases} \tag{65}$$

Additionally, in the limit, the top $k$ singular vectors of $\Sigma_{yx}$ acquire an alignment with the $k$ singular vectors the teacher network given by the relationship $|\langle \tilde{u}_i, \bar{u}_i\rangle||\langle\tilde{v}_i, \bar{v}_i\rangle| = \mathcal{O}(\bar{s}_i)$ where the the overlap function is defined as,

$$\mathcal{O}(\bar{s}_i) = \begin{cases} \left(1 - \frac{\mathcal{A}(1 + \bar{s}^2)}{\bar{s}(\mathcal{A} + \bar{s}^2)}\right)^{1/2}\left(1 - \frac{(\mathcal{A} + \bar{s}^2)}{\bar{s}(1 + \bar{s}^2)}\right)^{1/2} & \text{if } \bar{s} > \mathcal{A}^{1/4} \\ 0 & \text{otherwise} \end{cases} \tag{66}$$

# J    Experiment details

The codes to reproduce the experiments in the main paper can be found at `https://github.com/ccffccffcc/stochastic_collapse`. Our the experiments were run on the Google Cloud Platform (with $4\times$ or $8\times$ NVIDIA A100 (40GB) GPU). The initial code development occurred on a local cluster equipped with $10\times$ NVIDIA TITAN X GPUs. We carried out all the deep learning experiments with VGG-16 [47] and ResNet-18 [48], training on the CIFAR-10 and CIFAR-100 datasets respectively [49]. For VGG-16, we did not use Batch Normalization or Dropout. Both VGG-16 and ResNet-18 had their nonlinearity activation functions replaced with GELU, as the potentially larger invariant set associated with ReLU could complicate identification and illustration of the two main invariant sets discussed in the paper. We leave it as future work to study in details the structure of the invariant sets associated with ReLU. For all our training, we applied standard data augmentation and used SGD (with momentum $\beta = 0.9$ and weight decay of 0.0005) as the optimizer.

**Stochastic collapse in a quartic loss.** (Fig.1) The left three plot are generated by sampling 50 trajectories of the SDE $d\theta_t = -(\theta_t^3 - \mu\theta_t)dt + \zeta\theta_t dB_t$ with $\mu = 1.5$ and $\zeta = 0.1, 1, 2.5$. The samples are computed using an Euler-Maruyama discretization scheme with step size $\eta = 0.0025$ for $10^4$ steps. All three plots use the same random Gaussian initializations with mean zero and variance four. The side plot for these three subplots shows the theoretical steady-state distribution described in Sec. 4. The rightmost plot is generated by sweeping 50 linearly spaced values of $\zeta$ from $\zeta = 0$ to $\zeta = 3$. For each value of $\zeta$ we sample 10000 trajectories with the same scheme described previously but for a varying number of steps: $0, 5000, 10000, 20000, 40000, 80000$. We compute the empirical probability of the final position being within $\epsilon = 1e^{-15}$ euclidean distance of the origin.

**Stochastic collapse towards a sign invariant set.** (Fig.2) We trained $10^3$ different single-neuron models via simulating SDEs (31) up to time $T = 100$, using the Euler-Marumaya discretization scheme of step size $dt = 10^{-2}$. All the models are trained with the same data set, composed with 32 points $\{(x_i, y_i)\}_{i=1,\cdots,32}$ where $x_i$ is an i.i.d. standard Gaussian random variable, and $y_i$ is given by $y_i = x_i + \epsilon_i$ where $\epsilon_i$ is i.i.d. centered Gaussian random variable with standard deviation of 0.1. All the models are initialized with random weights sampled from i.i.d. centered Gaussian distribution with standard deviation $10^{-3}$.

**Evidence of stochastic collapse towards permutation invariant sets.** (Fig. 3) We trained VGG-16 for $10^5$ steps on CIFAR-10 with a learning rate of 0.1 and ResNet-18 for $10^6$ steps on CIFAR-100 with a learning rate of 0.02. We computed the normalized distance between neurons within the same layer by concatenating corresponding vectors of incoming and outgoing weights, and used this to cluster neurons. Neurons within 0.1 normalized distance from each other were grouped together, which then determines the number of clusters or the independent of neurons denoted as $n_{ind}$. In Fig. 3, we plotted the pairwise distance of the incoming weights and outgoing weights separately. We carried out Agglomerative Clustering based on the normalized distance of the incoming weights with $n_{ind}$ clusters and the corresponding pairwise distance between outgoing weights was visualized with the same ordering.

**Larger learning rates and increased label noise intensify stochastic collapse.** (Fig. 4 and 9) These experiments varied learning rates and the extent of label noise. For the learning rate experiments, we trained models without label noise for $10^5$ steps. For the label noise experiments, we maintained learning rates of 0.01 and 0.02 for VGG-16 and ResNet-18 respectively over $10^6$ steps. Here, fresh label noise was introduced with each batch sample. A label noise level of $\epsilon\%$ meant that $\epsilon\%$ of the labels were randomly assigned an incorrect label. We averaged the results across four replications with different seeds.

After training, we calculated and displayed the proportion of independent neurons in each layer of the networks. We first removed 'vanishing neurons', defined as those with incoming and outgoing weights less than 10% of the maximum norm for that layer. We then clustered the remaining neurons based on the pairwise normalized distance of the concatenated incoming and outgoing weights. Two neurons were defined as identical if their distance in weight space was less than 10% of their norms. This allowed us to determine the number of independent neurons in each layer.

**Effect of batch size on stochastic collapse.** (Fig. 10) We trained models with different batch sizes for $10^5$ (CIFAR-10) and $10^6$ (CIFAR-100) steps, while keeping the learning rate constant at 0.02. We averaged the results across four replications with different seeds. The method of calculating the proportion of independent neurons remains the same.

**Demonstrating generalization benefits of stochastic collapse in a teacher-student setting.** (Fig. 5) The experiments were conducted in accordance with assumptions A1 - A4. In these experiments, we set both the input and output dimensions to be 64, and the dataset size was set at 1024. We used a sparse teacher model with a rank of 8 and signals ranging evenly from 0.5 to 1. The input samples were drawn to satisfy the condition $X^T X = I$. We then constructed the signal $y$ by introducing a random Gaussian noise with a standard deviation of 0.5 to the true output values. The network was trained with an initial learning rate warm-up of 1000 steps, followed by 50000 steps at a learning rate of 3.0. After 50000 steps, we reduced the learning rate to 0.3 and continued the training for another 10000 steps. The experimental results are averaged over 256 runs with the same dataset.

In Fig. 11, we present the results obtained when replacing label noise with different batch sizes. In this experiment, we used SGD without label noise and maintained a similar experimental setup as before. Additionally, we introduced a learning rate drop at step 53000.

**Large learning rates aid generalization via stochastic collapse to simpler subnetworks.** (Fig. 6) We carried out initial training of VGG-16 and ResNet-18 using a larger learning rate of 0.1. We created checkpoints in the training process at steps of $10^4$, $2 \times 10^4$, and $4 \times 10^4$. We resumed training from the checkpoints post but at a reduced learning rate of 0.01. To track the impact of this large learning rate phase on the emergence of simpler subnetworks, we calculated the layer-wise fraction of independent neurons at each of these checkpoints, before the learning rate drop. We averaged the results across eight replications with different seeds.

# K Extended Experiments

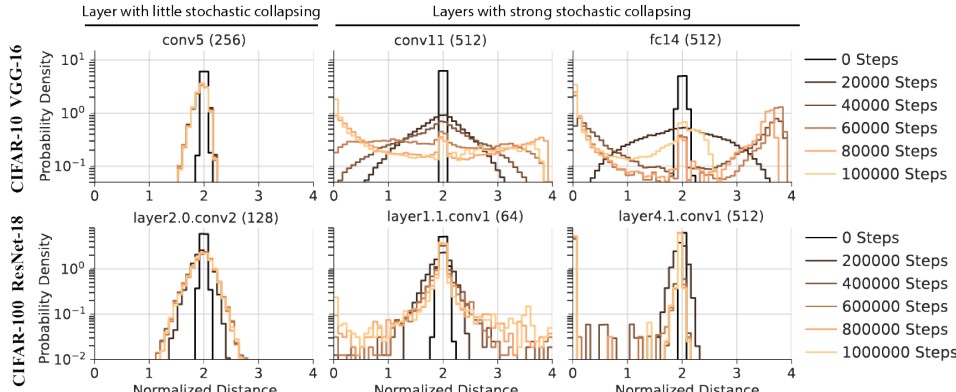

Figure 8: **Evidence of stochastic collapse towards permutation invariant sets in deep neural networks.** We show the distribution of normalized distance across all pairs of neurons for the same hidden layer of neurons as shown in Fig. 3. We show plots for three different layers in VGG-16 trained on CIFAR-10 (**top row**), and for three different layers in a ResNet-18 trained on CIFAR-100 (**bottom row**). Each plot assesses the distribution at various training stages. In layers with little stochastic collapsing (**left column**), the distributions rapidly thermalize and concentrate around 2. In layers with strong stochastic collapsing (**middle and right columns**), the distributions evolve to form a peak around 0, indicating stochastic collapsing towards the permutation invariant set.

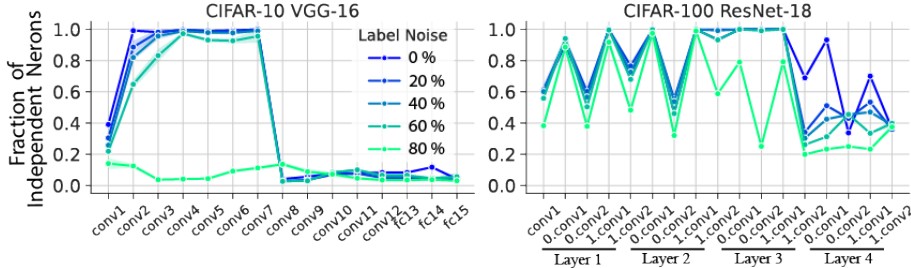

Figure 9: **Increased label noise intensify stochastic collapse.** This figure illustrates how the fraction of independent neurons per layer in VGG-16 trained on CIFAR-10 (**left column**) and ResNet-18 trained on CIFAR-100 (**right column**) varies with label noise. The networks are evaluated at training steps of $10^6$. A reduced percentage of independent neurons indicates stronger stochastic collapse.

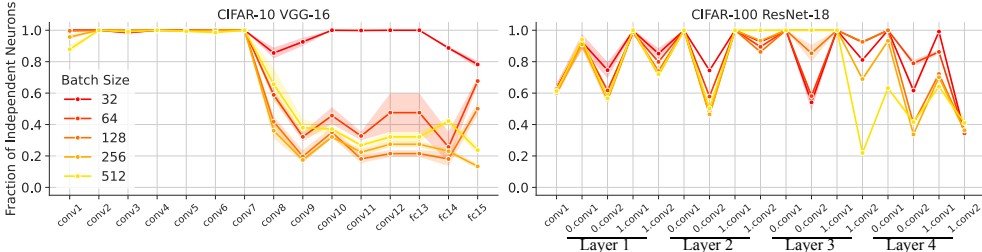

Figure 10: **Effect of batch size on stochastic collapse.** This figure illustrates how the fraction of independent neurons per layer in VGG-16 trained on CIFAR-10 (**left column**) and ResNet-18 trained on CIFAR-100 (**right column**) varies with batch size. A lower percentage of independent neurons signifies a heightened degree of stochastic collapse. As elaborated in Appendix D, modifying batch size is expected to parallel changes in the learning rate since both influence the amplitude term in the diffusion matrix. Yet, unlike learning rate effects, alterations in batch size produce complex shifts in stochastic collapse. We conjecture that this subtlety may arise from non-replacement sampling and the randomness introduced by transformations applied to input images.

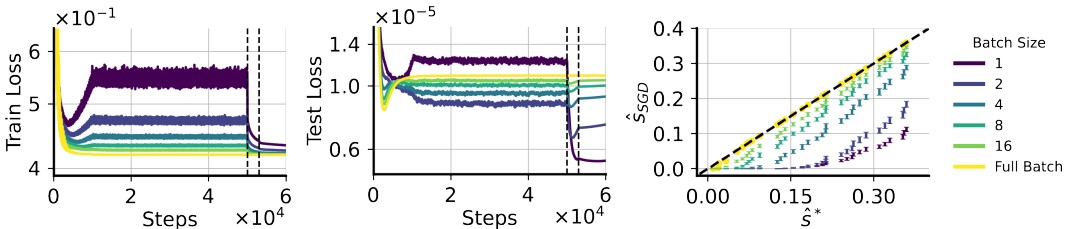

Figure 11: **Demonstrating generalization benefits of stochastic collapse in a teacher-student setting with batch noises.** Same as Fig. 5 but we show effects of different batch sizes. No label noise was used in this setup. We show the train loss (**left**) and test loss (**middle**) during the training of the student with different batch sizes. Dashed lines in the leftmost and middle left panels indicate the steps where learning rate is dropped. Training with smaller batch sizes generalizes better. **Right**: we show the noisy teacher signals against the learned student signals before dropping the learning rate. We only show the smallest 32 singular values.

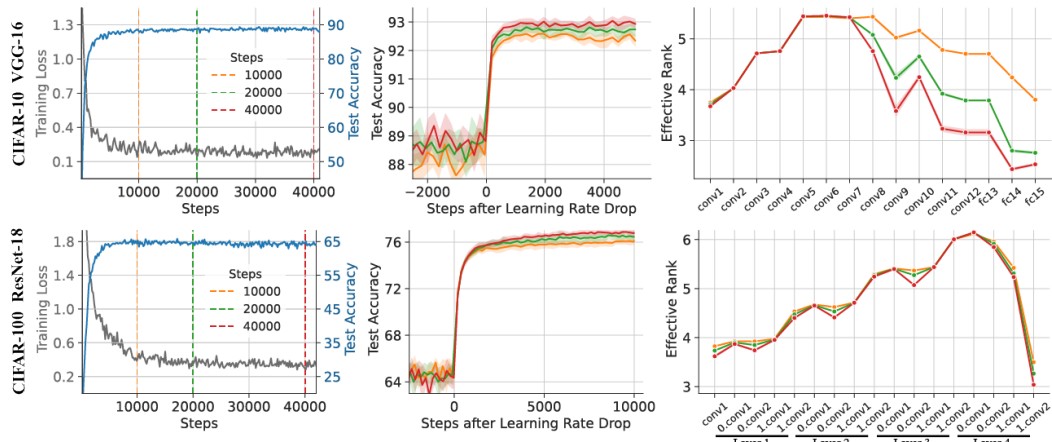

Figure 12: **Large learning rates aid generalization via stochastic collapse to simpler subnetworks.** **Left**, **Middle**: Same as Fig. 6. **Right**: The effective rank [60] of the weight matrices of independent neurons per layer evaluated at different learning rate drop times (indicated by the color) during the initial high learning rate training phase. To compute the effective rank, we first gather the singular values, denoted by $s$, of the concatenated weight matrices of the incoming and outgoing weights for a specified layer. The effective rank is then computed as $\rho = -\sum_i \hat{s}_i \log \hat{s}_i$, where $\hat{s}_i$ represents the normalized singular value defined by $\hat{s}_i = \frac{s_i}{\sum_j s_j}$.

