# OpenReview forum: "Stochastic Collapse: How Gradient Noise Attracts SGD Dynamics Towards Simpler Subnetworks"
_NeurIPS.cc/2023/Conference — NeurIPS 2023 poster_

### Official Review · Reviewer_DqpF · 2023-07-06

**Soundness:** 3 good
**Presentation:** 3 good
**Contribution:** 2 fair
**Rating:** 5
**Confidence:** 4

**Summary:**

The paper studies the properties of the stochastic gradient descent using its continuous time SDE approximation.   The invariance of the stochastic dynamics in the representation space are investigated and rigorous condition where such invariance sets acts as attractors of SDE are derived. Furthermore, it is empirically validated that neurons collapse towards these invariant sets and the generalization aspects of this phenomenon is studied in two-layer linear networks.

**Strengths:**

a) The paper tackles a very important question of understanding the regularization properties of stochastic algorithms. It formally characteries the two invariant properties of the stochastic algorithms on deep forward networks and established the conditions on the drift and variance terms of the SDE where these sets acts as attractors. In addition, in one-dimensional and two-dimensional settings the conditions are interpreted.

b) The above formalism established is novel and is useful to study the empirical behaviour of the deep networks and the role of the learning rate and its influence on generalization.

**Weaknesses:**

a)  The paper uses the label noise gradient descent and stochastic gradient descent interchangeably. However, note that there are two different algorithms, hence it should be explicitly stated for which algorithm the results are applicable. Although they share some similar properties, they also differ particularly in the strength of the noise Andriushchenko et al., Ziyin et. al.  The paper appears sloppy in this regard.

b) Although the paper establishes a more formal treatment the phenomenon of sign invariance and permutation invariance.  Andriushchenko et al. provides empirical invariance and theoretical intuitions on same phenomenon.



**Questions:**

a) In the results for evaluation of singular values of linear networks, the set of assumptions used appear strong. Particularly, the assumption that the assumption that noise in labels align with input-output covariance. Can it be verified that the noise of SGD satisfies this assumption  ?

b) The results in Figure 3 and Theorem 6.1 have been studied when label noise is introduced in the algorithm, how does these changed when using vanilla mini-batch SGD? The conclusion that large step size improve generalization is therefore cannot be generalized to mini-batch SGD.

**Limitations:**

The limitations are already addressed in the sections above.

---

> ### Author Rebuttal · Authors · 2023-08-09
>
> Thank you for your review and the suggestions to make our work better. We will discuss the weaknesses and questions of our paper you pointed out:
>
> **Weakness A**: We appreciate your valuable observation of the confusions that arose from our interchangeable use of noise introduced by minibatch SGD and label noise gradient descent (LNGD). This is important and we are committed to enhancing our paper in this regard. We firmly believe that the improvement we discuss below in each section will make our claims more solid:
> - Sec. 3: The existence of sign/permutation invariant sets is valid for both SGD and LNGD. Hence we can discuss these in the same scheme without any problem.
> - Sec. 4: Here, we argue that SGD can be seen as GD with a noisy term, leading us to study the stochastic gradient flow (SGF) as defined in Equation (4). Our theoretical analyses in Sec. 4 are firmly based on SGF. SGF can describe both SGD and LNGD, while their diffusion terms may have different structure.
> - Sec. 5,6: In these sections, we apply the theoretical results in Sec. 4 on SGF. We recognize that these sections need to be improved since we restrict ourselves to the case where the noise of SGF comes from the label noise, and hence we lose the generality of SGF. We used the label noise instead of the minibatch noise for the reason that label noise strength can be directly linked to the noise amplitude in SGF. However, in both these sections the analysis does not change if we let $\zeta$ be the mini-batch gradient noise amplitude. The other source of confusion might come from our empirical results, which use label noise (for the same reason discussed above). We now also include a new empirical figure (Fig. S3 in the attached PDF), in which we use SGD minibatch noises to validate the outcomes previously demonstrated solely with label noise (Fig. 3). Note that Fig. 1, the top of Fig. 2 and Fig. 4 do **not** use label noise, indicating strong stochastic collapse for minibatch SGD.
> - Appendix: In the appendix of our revised manuscript, we will add a section on the similarities and differences (both theoretically and empirically) between SGD and LNGD.
>
> We believe these improvements will greatly enhance our treatment of the subject and we hope they will positively influence your evaluation of our work.
>
> **Weakness B**: We thank the reviewer for pointing out the similarity between our analysis and the previous work by Andriushchenko et al., which we cited and discussed (lines 71-73) in our paper. Given the similarity to our work, we will add more discussions in our paper to address the substantial differences. The novelty of their work can be summarized as: 1. They revealed that large-step SGD dynamics have effective slow dynamics with multiplicative noise during loss stabilization. This allowed the authors to model the dynamics as a specific SDE, justified with empirics. 2. They theoretically analyzed diagonal linear networks, showing that the SDE has implicit bias toward sparser representations. 3. They conjectured that deep nonlinear networks also show this phenomenon, which is supported by the empirical results. 4. They argued that SGD first learns sparse features and then fits the data after the step size is decreased. With this foundation laid, we would like to emphasize two fundamental aspects that set our work apart:
>
> 1. Our analysis goes far beyond the basic understanding of diagonal linear networks, offering a broader perspective by introducing the invariant sets with a more general theorem (Theorem 4.1).  In their work, the authors conjectured an implicit bias towards sparser features beyond their diagonal linear model, leaving this exploration open as an "_exciting avenue for future work_". We believe that our paper makes substantial progress along this avenue by understanding the implicit bias towards sparse representations as stochastic collapse to the invariant sets.  Furthermore, our theory is applicable to general deep neural nets. We also provide a theoretical framework that predicts the quantitative conditions for this attractiveness in a general setting. While applying this theorem to arbitrary neural nets is a challenging future work, our work utilized this theorem to quantitatively analyze the collapsing threshold of simple models.
> 2. Our results shed light on the collapsing phenomena of weight vectors, a perspective that contrasts with their emphasis on neuron similarity based on activation patterns. Given that weight vector similarities often lead to similarity in activation patterns, our findings suggest that neural networks adhere to a stronger condition.
>
> **Question A**: We would like to point out that A1, A3, and A4 are standard assumptions used in the canonical papers [Saxe et al. 2014] and [Lampinen & Ganguli 2018], which establish the linear-teacher student model as a setting for studying generalization. As the reviewer rightly points out, the assumption that gradient noise aligns with input-output covariance (A2) is the new assumption we added such that we could derive analytical expressions for the theory when incorporating the new ingredient of stochasticity to the analysis. The technical difficulties for relaxing assumption A2 come from the problem that stochastic noises can perturb the eigenbasis, which adds an extra degree of complexity to the problem. Although we agree that relaxing those assumptions is theoretically interesting and important, we want to point out that our theory with those assumptions captures the key behavior of the general SGD dynamics. We now include additional empirical results (Fig. S3 in the attached PDF) which verifies that the key observed phenomena hold even without those assumptions and on minibatch SGD.
>
> **Question B**: The answer to this question is addressed in our response to Weakness A. In summary, the conclusion that large step sizes improve generalization does apply to mini-batch SGD without label noise (see Fig. 4 and Fig. S3 in the attached pdf).

---

> > ### Comment · Reviewer_DqpF · 2023-08-17
> > **Reply to Author Rebuttal**
> >
> > Thank you for your reply. My score remains unchanged. A few comments on the rebuttal below.
> >
> > Weakness A, Sec 5,6: Yes, I agree that you can set $\zeta$ to mini-batch noise amplitude however then it depends on (t) will not be constant and can be vanishing as the time progresses thus showing that collapse might not be stronger. It would be nice to make this point clear.

---

> > > ### Author Response · Authors · 2023-08-19
> > >
> > > Thank you for your reply and feedback, which has been very helpful in improving our work.
> > >
> > > The mini-batch noise should not depend on “t” directly, but rather implicitly through a dependency on position. The SDE used in our analysis in section 5 and 6 accounts for an aspect of this positional dependency through the multiplicative structure. That said, you are correct that there can be a further positional dependency unaccounted for in $\zeta$ when considering SGD instead of LNGD. For example, if an interpolating point exists, that is there exists some point $\theta_*$ where $\mathcal{L}(\theta_*) = 0$, then clearly the mini-batch noise amplitude near $\theta_*$ should be much smaller than at a random initialization.  One way to account for this would be to insert a dependency on the training loss $\mathcal{L}(\theta)$ into $\zeta$.  Unfortunately, this would make the analysis very complex and potentially intractable (we are currently working on if Theorem 5.1 can be extended in this manner). If the training data is not interpolatable, then there is always a positive lower-bound on $\zeta$, and we don’t expect the main lessons from this section would change drastically. As highlighted previously, the experiments for both these sections match well with the theory when run with SGD. This said, we should make it more clear what factors influence $\zeta$ in SGD and why we assume $\zeta$ is positionally independent in order to make the analysis tractable.
> > >
> > > We hope that the changes we discussed in our rebuttal and the above discussion alleviate your concerns around Weakness A in your review.  Have we addressed your other concerns and questions?  Is there an aspect of our work that is holding you back from raising your score?  Please let us know if you have any more questions.  We look forward to hearing from you.  And again, thank you for your time and feedback.

---

### Official Review · Reviewer_nG1u · 2023-07-08

**Soundness:** 3 good
**Presentation:** 3 good
**Contribution:** 2 fair
**Rating:** 5
**Confidence:** 4

**Summary:**

This paper demonstrate a low-dimensional invariant sets, namely a subset of parameter space, may remain unmodified by SGD. It means that SGD dynamics may lead to simple subnetworks. The theoretical mechanism behind is formally introduced. Moreover, the derived theoretical results revealed that the so-called stochastic collapse may influence generalization in a simplified linear teacher-student setup.

**Strengths:**

- The discovered stochastic collapse is interesting for understanding optimization dynamics of deep learning. Some counterintutive insights are provided to our community.

- Formal theoretical analysis with detailed proofs are presented for understanding stochastic collapse. It suggests that stochastic collapse can be precisely analyzed under certain assumptions.

- The empirical evidence is direct and informative.

**Weaknesses:**

- I think this paper needs to significantly improve the literature review and discussion. Some closely related studies (e.g. [1,2]) are absent. [1] demonstrated that SGD favor flat minima and SGD dynamics may happen in a low-dimensional parameter space. It will be helpful to explain how the simple subnetworks differs from and relates to the low-dimensional learning subspace in the previous studies. How close are the two low-dimensional subspaces? Will they provides some different insights?

- Generalization theoretical analysis is an important part of this paper. While the paper claimed that generalization benefits can be theoretical influenced by stochastic collapse, I did not see really novel or quantitative insights. If the theoretical results are really precise and informative, why not design some novel methods to improve generalization with theoretical supports? For example, if the role of early large learning rates can be well understood in the theory, it is possible to schedule the learning rate in a theory-guided manner that can work well in practice? If not possible, why?


Reference:

[1] Xie, Z., Sato, I., & Sugiyama, M. (2020, October). A Diffusion Theory For Deep Learning Dynamics: Stochastic Gradient Descent Exponentially Favors Flat Minima. In International Conference on Learning Representations.

[2] Gur-Ari, G., Roberts, D. A., & Dyer, E. (2018). Gradient descent happens in a tiny subspace. arXiv preprint arXiv:1812.04754.

**Questions:**

Please see the weaknesses.

**Limitations:**

I did see any potential negative societal impact.

The main limitation of this work lies its technical contributions, as I mentioned in the weaknesses.

---

> ### Author Rebuttal · Authors · 2023-08-09
>
> Thank you for your review and the time you spent suggesting ways to make our work better. We discuss the weaknesses of our paper you raised:
>
> **Weakness 1**: We appreciate the referenced works and will incorporate them into our study. We have updated our related work section to cite these works. We welcome any further suggestions on references or other concerns. Here we will focus specifically on these two works and their relationship to our work:
>
> Xie, Z., Sato, I., & Sugiyama, M.: We acknowledge the significance of this prior research, which connects the covariance matrix of SGD with the Hessian around local minima and  demonstrates that the escape rate from minima is linked to the Hessian eigenvalues or flatness. Furthermore, the authors briefly discussed that minima selection primarily occurs in a low-dimensional subspace spanned by the top Hessian eigenvectors. Our works both expands and diverges from this analysis in a couple of key ways:
>
> 1. New concepts: We introduce the concept of sign/permutation invariant sets, bridging low-dimensionality of SGD dynamics with the sparsity of the network representations. In the vicinity of the invariant sets, the Heissan along orthogonal directions has near-zero eigenvalues, and hence we believe that the low-dimensional subspace discussed by Xie, Z. et al. is nested in one of these invariant sets we introduced. Verifying this hypothesis is interesting future work.
>
> 2. Distinct Mechanism: We reveal the mechanism on why the low-dimensional invariant sets can be attractive. This mechanism is distinct from the previous analyses for two reasons. First, their work is based on the Kramers Escape Problem. Contrarily, in our investigation, the stochastic attractivity comes from the multiplicative nature of the noise and the effective entropy of invariant sets, which means that the escape rate doesn't bear direct relevance in our context. Second, while the paper briefly discussed SGD's biases towards low-dimensional subspaces, their analyses are built on the empirical fact that the Hessian has many near-zero eigenvalues with a few large ones. The theoretical explanation of this empirical fact is not provided. In contrast, our analyses are not tied to specific assumptions about the Hessian spectrum, marking a different analytical perspective.
>
> Gur-Ari, G., Roberts, D. A., & Dyer, E: This insightful work shows that the Hessian splits into two slowly varying subspaces during the course of training and the gradient lives in the subspace spanned by the principal eigenvectors. They also highlight the exploration of the "_transparent meaning_" of these eigenvectors for future research. In line with this, our investigation suggests a potential correlation between the low-dimensional space and the invariant sets. We consider this connection a promising area for exploration and believe it could contribute to understanding the "_transparent meaning_" that the previous work hinted at.
>
> **Weakness 2**: We agree that designing improved algorithms based on our insights into stochastic collapse is a major goal for our future work. Nonetheless, the goal of this work was to highlight a novel, not yet well understood (and indeed undiscovered) implicit bias of SGD, stemming from the interplay between stochasticity and sparsity. This said, we are actively developing learning rate schedules that use the understanding of stochastic collapse defined and presented in this work.  We share some additional thoughts on this. As detailed at the end of Section 6 (lines 312-328) and Fig. 4, we found that a prolonged initial training phase with larger learning rates are beneficial for subsequent generalization due to a bias toward sparser representations. Inspired by our theory, we empirically counted the number of independent neurons as a measure of sparsity. We conjecture that this is a good indicator for when to drop down the learning rates. The prolonged training with larger learning rates is beneficial as long as it continuously reduces the network sparsity. One idea for the algorithm is to measure the number of independent neurons (can be done sporadically to reduce computational costs) and drop down the learning rates when this number plateaus. One technical challenge here is to define a single variable to quantify the entire network's simplicity rather than layer-by-layer. This challenge, along with benchmarking the algorithm across various datasets, demands substantial effort and extends beyond this paper's scope. Nevertheless, we recognize the significance of such algorithmic design and embrace it as an essential future endeavor, made possible by our theory/empirics!
>
> Besides, we would like to point out that our work contributes novel and quantitative insights into the effect of stochastic collapse on generalization, explored within a linear teacher-student setting.  Our analysis highlights an interesting tradeoff between maintaining enough gradient noise to induce stochastic collapse in directions with weak signal-to-noise ratios, while also reducing this noise to facilitate annealing in directions with robust signal-to-noise ratios. Also, it reveals the surprising finding that SGD can get attracted to **higher** training error saddle points (student with some zero singular values) that nevertheless have **lower** test error than local minima (student with all nonzero singular values) with lower training error.
>
> In summary, we believe our paper provides both novel theoretical insights and a foundation for future practical applications. While we recognize the value and necessity of translating these insights into algorithmic designs, the work presented here primarily serves as a theoretical/scientific experiment work. Your feedback has been instrumental in refining our focus, and we hope this response successfully addresses your queries.

---

> > ### Comment · Reviewer_nG1u · 2023-08-14
> > **Thanks for the response.**
> >
> > Thanks for the response.
> >
> > It helps address some of my concerns.
> >
> > I plan to raise the score at this point.

---

> > > ### Author Response · Authors · 2023-08-14
> > >
> > > We are happy to hear that our response resolved some of your concerns and we are glad to know you plan on raising your score. Please let us know if you have any questions.

---

> > > > ### Author Response · Authors · 2023-08-20
> > > >
> > > > Thank you for considering our response. Please let us know if you have any questions in order to determine the score you plan to raise to before the discussion phase ends on Aug. 21st. Thank you!

---

### Official Review · Reviewer_kjwn · 2023-07-09

**Soundness:** 3 good
**Presentation:** 3 good
**Contribution:** 3 good
**Rating:** 6
**Confidence:** 2

**Summary:**

This paper studies the implicit bias of SGD, and provides a new perspective by characterizing the invariant set in the parameter space along the SGD dynamics, corresponding to specific model architectures. It is shown that SGD noise induces stochastic attractivity towards such invariant sets, which leads to simpler subnetworks. Such simplicity bias is also shown to be beneficial for generalization. Empirical evidence is also provided.

**Strengths:**

This paper provides a novel perspective for characterizing the implicit bias of SGD. The empirical and theoretical analyses in the current paper seem interesting and insightful.

**Weaknesses:**

1. The proof of the stochastic attractivity seems to only apply to very simple networks. For sign invariance, Theorem 5.1 applies to a scalar single neuron model; for permutation invariance, only a two-neuron network is analyzed.
2. Similarly, analysis of the training dynamics only applies to two-layer linear networks. These seem insufficient for justifying what is really happening during actual training of deep neural networks.

**Questions:**

1. The invariant sets for SGD are also invariant sets for GD. Is there evidence that GD doesn't exhibit collapse?
2. Is there any interplay between different types of invariant sets? E.g., which type of invariance does SGD prefer?
3. Is it possible to characterize an invariance type by a corresponding norm of the parameters? If not, why?
4. In general, how to verify the sufficient conditions in Theorem 4.1? Is it possible to verify them empirically for those models used in the experiments in the current paper?

**Limitations:**

The authors have adequately addressed the limitations.

---

> ### Author Rebuttal · Authors · 2023-08-09
>
> Thank you for your effort on the insightful review and suggestions. We are going to discuss each weaknesses and questions raised:
>
> **Weakness 1**: We value your insight but respectfully disagree.  In Section 4 of our paper, Theorem 4.1 yields sufficient conditions for stochastic collapse applicable to **any** affine invariant set of **any** dimension in **any** network.  We acknowledge that stronger statements in Sec. 5 & 6 are restricted to simpler settings, partially to provide concrete/intuitive examples to illustrate our general theory.  Deriving sufficient and necessary conditions for stochastic collapse for all neural network architectures lies beyond present theory, as we explain in our revised section 4:
>
> “_To extend the collapsing condition derived in Theorem 4.2 to high-dimensional cases, a natural idea would be to consider all one-dimensional slices of parameter space orthogonal to the invariant set. However, the issue is that some of these slices might satisfy the collapsing condition while others do not. This can result in complex dynamics near the boundary of the invariant set making it difficult to derive a necessary and sufficient condition for attractivity in high-dimensions. Nonetheless, we can derive a sufficient condition._”
>
> **Weakness 2**: We would like to clarify the goal of Sec. 6. It is a fundamental challenge to derive analytic learning dynamics for general deep nets. Sec. 6 provides an instructive setting for understanding generalization in deep learning dynamics where we could derive exact results on how stochastic collapse impacts generalization. We used this understanding to derive **new** testable predictions about the role of stochastic collapse in non-linear models. As explained in the paragraph following the introduction of the linear-teacher student analysis (lines 312 - 319):
>
> “_The key prediction is that a large learning rate induces stronger stochastic collapse, thereby regularizing the model complexity. Furthermore, remaining in a phase of larger learning rates for a prolonged period drives SGD closer to the invariant set. Consequently, when the learning rate is eventually dropped, overfitting in these specific directions is mitigated._”
>
> In the subsequent paragraph, we proceeded to “_test this predicted mechanism_” on a “_VGG-16 and ResNet-16 [trained] on CIFAR-10 and CIFAR-100, respectively_”. And as demonstrated in figure 4, “_we found, as predicted, that models with a longer initial training phase collapse towards simpler invariant sets with fewer independent neurons_”.
>
> This combination of deriving a theoretical prediction from analytic insights into a simple model, and then testing this prediction in more complex models, that are currently beyond the reach of theory, is a powerful route forward for advancing our conceptual understanding of the complex processes of deep learning; it combines theory and experiment in creative/synergistic ways, obtaining results that neither theory nor experiment alone could achieve.  Therefore this connection between the linear student-teacher setting and more complex models should, we believe, be considered a strength of our work rather than a weakness.  We never would have tested a connection between duration of large learning rate training and degree of collapse in Fig. 4 without our theory.  We also would never have tested for strong stochastic collapse to permutation invariant sets in Fig. 1 & 2 without our theory.
>
> **Question 1**: While invariant sets of SGD are also invariant sets of GD, such sets can be attractive for SGD but **not** for GD.  Our surprising result is that if the diffusion of SGD is strong enough, SGD can be attracted to invariant sets that **repel** GD (examples: local maximum of double well potential in Sec. 4, saddle points with zero singular values in linear-student teacher setting in Sec. 6). Theorem 4.1 & 4.2 shows precisely how diffusion is essential for collapse of SGD to maxima/saddles; GD cannot similarly collapse because diffusion is absent; GD can only collapse to minima.
>
> **Question 2**:  Here are factors that we believe influence SGD’s likelihood of collapse:
> - Attractivity Rate: SGD is more inclined towards invariant sets with a higher attractivity rate, a key quantity defined in Theorem 4.2.
> - Quantity of Invariant Sets: SGD favors more numerous invariant sets; e.g. the number of permutation invariant sets within each layer of $n$ neurons is $O(n^2)$ while the number of sign invariant sets is only $O(n)$. This suggests collapse to permutation invariant sets is more common (see Fig. 1).
>
> **Question 3**: We did characterize invariant types by a norm of the parameters:
> - Sign Invariant Set: For a given neuron, its sign invariant set is defined by the vanishing norm of both its incoming and outgoing weight vectors (lines 235 - 236).
> - Permutation Invariant Set: For two neurons, their permutation invariant set is determined by the vanishing norm of the difference between their incoming and outgoing weight vectors (lines 227 - 229 and lines 236 - 239).
>
> **Question 4**: Verifying the sufficient condition of theorem 4.1 in deep nets is technically hard because: (1) we must check equation (7) holds for any unit normal vector and every point $\theta$ in the invariant set; (2) we must compute the full Hessian matrix and second derivative of the diffusion matrix, within the entire invariant set, which is expensive.  Although we cannot directly verify theorem 4.1 for modern neural nets, we have verified it in simpler examples. We now include additional figures (see Figure S1 and S2 in the attached PDF) for the verification of Theorem 4.2 and 5.1, which thus also verifies Theorem 4.1.  But **most importantly** to circumvent these difficulties, we **directly** demonstrate stochastic collapse in realistic settings of ResNets and VGG (Fig. 1,2,4); this **direct** empirics constitutes a powerful demonstration of the prevalence of stochastic collapse in practice.

---

> > ### Comment · Reviewer_kjwn · 2023-08-16
> > **Response to the authors**
> >
> > I thank the authors for the detailed response. I don't have further questions, and I'm willing to raise the score.

---

> > > ### Author Response · Authors · 2023-08-17
> > >
> > > Thank you for your consideration in revising the score! Please let us know if any additional questions come up.

---

### Official Review · Reviewer_C15e · 2023-07-12

**Soundness:** 3 good
**Presentation:** 3 good
**Contribution:** 3 good
**Rating:** 6
**Confidence:** 2

**Summary:**

The paper explores how stochastic gradient descent (SGD), a common optimization method for deep neural networks, can lead to simpler and more generalizable models. The paper introduces the concept of invariant sets, which are regions of the parameter space that are unaffected by SGD, and shows how SGD can be attracted to these sets under certain conditions. The paper also analyzes how this attraction can result in removing or reducing redundant neurons in the network, and how this can improve the performance on unseen data. The paper provides theoretical and empirical evidence for these phenomena, and explains why training with large learning rates for a long time can help SGD find simpler subnetworks.

**Strengths:**

- The paper is well-motivated, focusing on the study of how gradient noise directs SGD dynamics toward simpler subnetworks.
- Several interesting insights are presented, including the relationship between learning rate and stochastic collapse, and the influence of the stochastic collapse on generalization.

**Weaknesses:**

- The paper does not sufficiently clarify the applicability of the attractivity condition to permutation invariant sets in high-dimensional settings. Additionally, it inadequately addresses the extension of the linear teacher-student framework to non-linear models or more general scenarios.

**Questions:**

- Is it possible to (theoretically) analyze stochastic collapse beyond linear models or one-dimensional settings?

**Limitations:**

No potential negative societal impact.

---

> ### Author Rebuttal · Authors · 2023-08-09
>
> We appreciate your comprehensive review and the time you have taken to suggest improvements for our study. We'd like to respond to the weaknesses and queries you raised regarding our paper:
>
> **Weakness 1**: “_not sufficiently clarifi[ing] the applicability of the attractivity condition to permutation invariant sets in high-dimensional settings._” While we did not provide a direct theoretical treatment of our theorems in section 4 to a high-dimensional scenario of the permutation invariant set (while we did do this for the sign invariant set in the teacher-student analysis in section 6). As we explained on lines 221-223, “_we provide intuitive insights into the attractivity of permutation invariant sets through a toy example of a two-neuron neural network in Appendix D_”.  We attempted to analyze this setting theoretically with high-dimensional input/output, but found it very difficult to derive statements stronger than what the main theorem statements in section 4 already implied. Note that our main Theorem 4.1 provides a sufficient condition for stochastic attractivity to **any** affine invariant set, **including** permutation invariant sets.  To further bolster this result, we devoted the majority of section 5 in our paper to empirically verifying and exploring the applicability of our attractivity condition to permutation invariant sets in high-dimensional neural network settings.  We found, remarkably, strong stochastic collapse to permutation invariant sets in which many neurons have identical incoming and outgoing weights (Fig. 1) in realistic networks (ResNets and VGG-16).  We believe our combination of theory and empirics provides strong applicability of the phenomenon of  attraction to permutation invariant sets.
>
> **Weakness 2**: “_extension of the linear teacher-student framework to non-linear models or more general scenarios._” We appreciate your comment and would like to clarify how we extended the linear teacher-student framework to the more realistic neural network settings.  Directly extending our theoretical analysis to more complex non-linear models is exceptionally challenging, demanding the introduction of novel analysis techniques, and would stand as a distinct and independent research endeavor. Thus, the goal of section 6 was to use a well-studied setting for understanding generalization in deep learning dynamics where we could derive exact results highlighting the influence of stochastic collapse on generalization.  We used this understanding to derive testable predictions about the role of stochastic collapse in non-linear models. As explained in the paragraph following the introduction of the linear-teacher student analysis (lines 312 - 319):
>
> “_The analyses in the linear teacher student network provide valuable insights into how stochastic collapsing can enhance generalization…The key prediction is that a large learning rate induces stronger stochastic collapse, thereby regularizing the model complexity. Furthermore, remaining in a phase of larger learning rates for a prolonged period drives SGD closer to the invariant set. Consequently, when the learning rate is eventually dropped, overfitting in these specific directions is mitigated._”
>
> In the subsequent paragraph, we proceeded to “_test this predicted mechanism_” on a “_VGG-16 and ResNet-16 [trained] on CIFAR-10 and CIFAR-100, respectively_”. And as demonstrated in figure 4, “_we found, as predicted, that models with a longer initial training phase collapse towards simpler invariant sets with fewer independent neurons_”.
>
> This combination of deriving a theoretical prediction from analytic insights into a simple model, and then testing this prediction in more complex models, that are currently beyond the reach of theory, is a powerful route forward for advancing our conceptual understanding of the complex processes of deep learning; it combines theory and experiment in creative, powerful and synergistic ways, obtaining results that neither theory nor experiment alone could achieve.  Therefore this connection between the linear student-teacher setting and more complex models should, we believe, be considered a strength of our work rather than a weakness.
>
> Answer to the question: Yes, our work **does** indeed analyze stochastic collapse in non-linear or high-dimensional setups. Specifically, Section 4 aims to offer a general theorem for stochastic collapse within a broader framework. Following Section 4, we study stochastic collapse in a **non-linear** but single-neuron model (Theorem 5.1), while our teacher-student framework in Section 6 studies stochastic collapse in a **high-dimensional** linear model. We agree that finding a non-linear **and** high-dimensional setting where studying stochastic collapse is theoretically tractable is an important and challenging step for future work. Nonetheless, our current work fills in this theoretical gap by exploring these settings through extensive empirics.  We verified empirically the prevalence and importance for generalization of  stochastic collapse in realistic networks (ResNets and VGG) (Fig. 1,2, and 4), and demonstrate that stochastic collapse in these realistic settings shows similar properties to those we have found in the simple examples.

---

> > ### Comment · Reviewer_C15e · 2023-08-18
> >
> > Thanks for your helpful responses. The reviewer has no further questions and will keep the positive rating of the paper.

---

> > > ### Author Response · Authors · 2023-08-19
> > >
> > > Thank you for your reply.  We are happy to hear you think positively about the paper.  Is there an aspect of our work that is holding you back from raising your score?  Your comments will help us improve our paper further. Please let us know if you any new questions do come up.

---

> > > > ### Comment · Reviewer_C15e · 2023-08-19
> > > >
> > > > While the paper does contain some restrictive assumptions, it also provides some interesting insights, the reviewer appreciates the value of this work and plans to increase the score from 5 to 6.

---

> > > > > ### Author Response · Authors · 2023-08-20
> > > > >
> > > > > We are happy to hear you appreciate the value of our work and believe that it provides interesting insights.  Thank you for revising your score!

---

### Official Review · Reviewer_ZL9R · 2023-07-30

**Soundness:** 4 excellent
**Presentation:** 4 excellent
**Contribution:** 3 good
**Rating:** 7
**Confidence:** 3

**Summary:**

The authors investigate the implicit bias of the SGD algorithm towards invariant sets -- with a focus on sign and permutation invariance. In particular, they derive a sufficient condition for the stochastic attraction of the SDE describing the first two moments of the SGD process towards these invariant sets, a phenomenon that they call "stochastic collapse". Then, they turn to the study of generalization error dynamics in a one-hidden layer teacher-student linear network, showing that large learning rates increase stochastic collapse, that in turn benefits generalization by implicit low-rank regularization. Finally, they test their theoretical predictions on realistic architectures.

**Strengths:**

The paper presents an interesting perspective on the implicit bias of SGD and its beneficial effect on generalization, by bridging the gap between different approaches -- e.g., the interplay of the Hessian and the effective noise, the tuning of the learning rate, the role of sparsity. The results are solid and well presented.

**Weaknesses:**

- The results are obtained only in continuous time
- Assumptions (A1-A4) in Sec. 6 are really restrictive


**Questions:**

- On line 176, the authors comment that stochastic collapse can bias the SGD dynamics towards local maxima of the loss. How can this result in better generalization? A comment in relation to Sec. 6 would be useful.

**Limitations:**

Limitations have been adequately addressed.

---

> ### Author Rebuttal · Authors · 2023-08-09
>
> Thank you for your insightful review and the time you took to identify areas where our work could be improved. We'd like to address the concerns you raised:
>
> **Weakness A**: We appreciate your observation regarding our use of a continuous formulation of SGD and we agree that this is a limitation of our current analysis. As we discussed at the end of our manuscript, “_extending our analytic results from continuous SGF to the discrete SGD updates_” is an important direction for future work. Our decision to adopt a continuous formulation was driven by our desire to leverage existing concepts from stochastic control theory, such as the definition of stochastic attractivity and the use of tools such as Itô's calculus to derive concise conditions for stochastic attractivity. However, we'd like to underline that our paper's main arguments retain their strength for several reasons:
> - Our discussion of the invariance properties of sign-invariant and permutation-invariant sets (refer to Lemma 4.1 in our paper) is not confined to the continuous formulation; these properties remain valid in the original discrete SGD setting as well.
> - Our empirical examinations with CNN models utilizes the conventional discrete SGD algorithm instead of a discretization of the continuous SGF variant discussed in our theoretical framework. The consistency of our empirical findings with our theoretical postulates underscores that our continuous assumption is sufficient to explain important features of the outcome of discrete SGD updates at practical finite learning rates typically used.
>
> In our updated manuscript we will add a discussion further elaborating on our choice to use  a continuous formulation of SGD and the approach to extend our findings to the discrete scenario.
>
> **Weakness B**: We want to emphasize that assumptions A1, A3, and A4 are standard assumptions used in the canonical papers [Saxe et al. 2014] and [Lampinen & Ganguli 2018], which established the linear-teacher student model as a setting for studying generalization. We added A2 such that we could derive analytical expressions for the theory when incorporating the new ingredient of stochasticity to the analysis. We agree that it is important to check how restrictive these assumptions are and we thank the reviewer for pointing this out. We now include additional empirical results (see Figure S3 in the attached PDF) which verifies that the key observed phenomena hold even without these assumptions. Although we acknowledge that our assumptions in section 6 are restrictive, we believe they are necessary to provide an analytic demonstration of how stochastic collapse influences generalization, though the derived conclusions hold more generally. Indeed remarkably, our theory, despite its restrictions made a **new** prediction that training for longer at larger learning rates helps generalization because it promotes stochastic collapse.  We tested this novel prediction successfully in regimes far beyond the restrictions of our theory  - in ResNets and VGG’s trained on CIFAR10/100 (see Fig. 4).  Thus our simple theory makes powerful, generally applicable predictions.
>
>
> **Answer to question for line 176**: We appreciate your inquiry. Upon revisiting this sentence, we agree that it is somewhat ambiguous and might give the impression that converging to a local maxima is beneficial. Although there should exist certain pathological instances where such convergence is beneficial,  in general this would not be the case. For a high-dimensional setting, such as that considered in section 6, stochastic collapse to saddle points, not local maxima, would be beneficial to generalization. The sentence should read, "_converge to a local maxima or **saddle-point** of the loss landscape_".  For example, in our linear student-teacher setting, fixed points where the student has some of its singular values stochastically collapse to zero (corresponding to small data singular values that are not learned), are actually saddle points in the training error loss landscape, and these saddle points have **higher** training error than the global minimum (in which all the data singular values are learned for a full rank student), but **lower** test error than the global minimum.  This is thus a concrete example of how convergence to a higher training error saddle point can, remarkably, help generalization.  We will expand upon this important loss landscape perspective in the revised camera-ready paper.

---

> > ### Comment · Reviewer_ZL9R · 2023-08-14
> >
> > I thank the authors for their careful responses. My concerns have been adequately address and I remain of the opinion that this work is interesting and worth publication. Therefore, I will keep my score of 7.

---

> > > ### Author Response · Authors · 2023-08-14
> > >
> > > Thank you for your continued support.  Let us know if additional questions come up.

---

### Author Rebuttal · Authors · 2023-08-09

We would like to thank all reviewers for their careful and detailed comments. We here attach a single-page pdf with the following three figures to address reviewers’ questions.

1. Figure S1-  Empirics for our illustrative example. This shows that the quantitative prediction via Theorem 4.2 agrees with empirical results.
2. Figure S2 - Empirics for the single neuron model in Section 5. This shows that the quantitative prediction via Theorem 5.1 agrees with empirics results
3. Figure S3 - Empirics for the teacher-student setup in Section 6, where we verify that the key observed phenomena shown in Figure 3 hold even without the assumptions proposed in Section 6 and on SGD minibatch noises.

Please also see our individual responses to each reviewer for more information!

Lastly, it's worth pointing out that a crucial conceptual advance within our paper, though present, hasn't been prominently emphasized. This insightful perspective is as follows: **Anisotropic position dependent diffusion can cause SGD in deep learning to get attracted to higher training error saddle points that nevertheless have lower test error than local minima with lower training error.**  This statement is very surprising to many people we have discussed with, yet it is nevertheless true and our paper explains why.  For example, in our linear student-teacher setting, fixed points where the student has some of its singular values stochastically collapse to zero (corresponding to small data singular values that are not learned), are actually saddle points in the training error loss landscape, and these saddle points have **higher** training error than the global minimum (in which all the data singular values are learned for a full rank student), but **lower** test error than the global minimum (also, more generally, any network at a learning fixed point with identical neurons typically is at a saddle; lower training error can be achieved by specializing the neurons).  This is thus a concrete example of how convergence to a higher training error saddle point can, remarkably, help generalization.  We believe this may become an impactful,  well-known conceptual advance in deep learning due to our paper. We will expand upon this important loss landscape perspective in the revised camera-ready paper.

We greatly appreciate the time and effort you spent reviewing our paper, which we are certain will make our work better. We hope you will consider our paper an important contribution to the NeurIPS community. We look forward to hearing from you!

---

### Decision · Program_Chairs · 2023-09-21

**Decision:**

Accept (poster)

**Comment:**

This work characterizes the attractivity of stochastic gradient flow to some invariant sets. The paper considers different types of invariant sets such as sign invariant sets and permutation invariant sets, which correspond to sparse or low-rank sub-networks. They show that SGF ends up in these invariant sets during training and that these invariant sets are connected to better generalization.

The reviewers raised a number of concerns, most of which have been successfully addressed in the rebuttal, including the restriction to SGF (SGF is often used as a model to SGD) or the generality of the results (stronger statements in Sec. 5 & 6 are restricted to simpler settings). Another important concern raised by some reviewers is the lack of discussion of prior work. For instance, there is a very large body of literature on double-well potentials or Kramer's escape problems. Perhaps a good reference to start with is https://hal.science/hal-00604399v2/document. Finally, there is a lot of recent work that studies the diffusion properties of SGF, including some references mentioned by the reviewers. I would encourage the authors to expand the related work section to connect their work to these prior results.

Overall, the paper makes some nice contributions but it suffers a bit from a lack of clarity in the writing, which perhaps explains why the scores it receives are on the borderline accept side. I would encourage the authors to try to distill their main messages in a better way if possible. That said, I think this does not require a major revision and would therefore happily accept this paper.